# SkillS: Adaptive Skill Sequencing for Efficient Temporally-Extended Exploration

## Abstract

The ability to effectively reuse prior knowledge is a key requirement when building general and flexible Reinforcement Learning (RL) agents. Skill reuse is one of the most common approaches, but current methods have considerable limitations. For example, fine-tuning an existing policy frequently fails, as the policy can degrade rapidly early in training. In a similar vein, distillation of expert behavior can lead to poor results when given sub-optimal experts. We compare several common approaches for skill transfer on multiple domains including changes in task and system dynamics. We identify how existing methods can fail and introduce an alternative approach to mitigate these problems. Our approach learns to sequence existing temporally-extended skills for exploration but learns the final policy directly from the raw experience. This conceptual split enables rapid adaptation and thus efficient data collection but without constraining the final solution. It significantly outperforms many classical methods across a suite of evaluation tasks and we use a broad set of ablations to highlight the importance of different components of our method.

## 1 Introduction

The ability to effectively build on previous knowledge, and efficiently adapt to new tasks or conditions, remains a crucial problem in Reinforcement Learning (RL). It is particularly important in domains like robotics where data collection is expensive and where we often expend considerable human effort designing reward functions that allow for efficient learning.

Transferring previously learned behaviors as skills can decrease learning time through better exploration and credit assignment, and enables the use of easier-to-design (e.g. sparse) rewards. As a result, skill transfer has developed into an area of active research, but existing methods remain limited in several ways. For instance, fine-tuning the parameters of a policy representing an existing skill is conceptually simple. However, allowing the parameters to change freely can lead to a catastrophic degradation of the behavior early in learning, especially in settings with sparse rewards (Igl et al., 2020). An alternative class of approaches focuses on transfer via training objectives such as regularisation towards the previous skill. These approaches have been successful in various settings (Ross et al., 2011; Galashov et al., 2019; Tirumala et al., 2020; Rana et al., 2021), but their performance is strongly dependent on hyperparameters such as the strength of regularisation. If the regularisation is too weak, the skills may not transfer. If it is too strong, learning may not be able to deviate from the transferred skill. Finally, Hierarchical Reinforcement Learning (HRL) allows the composition of existing skills via a learned high-level controllers, sometimes at a coarser temporal abstraction (Sutton et al., 1999). Constraining the space of behaviors of the policy to that achievable with existing skills can dramatically improve exploration (Nachum et al., 2019) but it can also lead to sub-optimal learning results if the skills or level of temporal abstraction are unsuitably chosen (Sutton et al., 1999; Wulfmeier et al., 2021). As we will show later (see Section 5), these approaches demonstrably fail to learn in many transfer settings.

Across all of these mechanisms for skill reuse we find a shared set of desiderata. In particular, an efficient method for skill transfer should 1) reuse skills and utilise them for exploration at coarser temporal abstraction, 2) not be constrained by the quality of these skills or the used temporal abstraction, 3) prevent early catastrophic forgetting of knowledge that could be useful later in learning.

With Skill Scheduler (SkillS), we develop a method to satisfy these desiderata. We focus on the transfer of skills via their generated experience inspired by the Collect & Infer perspective (Ried-

miller et al., 2022; 2018). Our approach takes advantage of hierarchical architectures with pretrained skills to achieve effective exploration via fast composition, but allows the final solution to deviate from the prior skills. More specifically the approach learns two separate components: First, we learn a high-level controller, which we refer to as scheduler, which learns to sequence existing skills, choosing which skill to execute and for how long. The prelearned skills and their temporally extended execution lead to effective exploration. The scheduler is further trained to maximize task reward, incentivizing it to rapidly collect task-relevant data. Second, we distill a new policy, or skill, directly from the experience gathered by the scheduler. This policy is trained off-policy with the same objective and in parallel with the scheduler. Whereas the pretrained skills in the scheduler are fixed to avoid degradation of the prior behaviors, the new skill is unconstrained and can thus fully adapt to the task at hand. This addition improves over the common use of reloaded policies in hierarchical agents such as the options framework (Sutton et al., 1999) and prevents the high-level controller from being constrained by the reloaded skills.

The key contributions of this work are the following:

- We propose a method to fulfil the desiderata for using skills for knowledge transfer, and evaluate it on a range of embodied settings in which we may have related skills and need to transfer in a data efficient manner.
- We compare our approach to transfer via vanilla fine-tuning, hierarchical methods, and imitation-based methods like DAGGER and KL-regularisation. Our method consistently performs best across all tasks.
- In additional ablations, we disentangle the importance of various components: temporal abstraction for exploration; distilling the final solution into the new skill; and other aspects.

## 2 RELATED WORK

The study of skills in RL is an active research topic that has been studied for some time (Thrun & Schwartz, 1994; Bowling & Veloso, 1998; Bernstein, 1999; Pickett & Barto, 2002). A 'skill' in this context refers to any mapping from states to actions that could aid in the learning of new tasks. These could be pre-defined motor primitives (Schaal et al., 2005; Mülling et al., 2013; Ijspeert et al., 2013; Paraschos et al., 2013; Lioutikov et al., 2015; Paraschos et al., 2018), temporally-correlated behaviors inferred from data (Niekum & Barto, 2011; Ranchod et al., 2015; Krüger et al., 2016; Lioutikov et al., 2017; Shiarlis et al., 2018; Kipf et al., 2019; Merel et al., 2019; Shankar et al., 2019; Tanneberg et al., 2021) or policies learnt in a multi-task setting (Heess et al., 2016; Hausman et al., 2018; Riedmiller et al., 2018). It is important to note that our focus in this work is not on learning skills but instead on how to to best leverage a given set of skills for transfer.

Broadly speaking, we can categorise the landscape of transferring knowledge in RL via skills into a set of classes (as illustrated in Fig. 1): direct reuse of parameters such as via fine-tuning existing policies (Rusu et al., 2015; Parisotto et al., 2015; Schmitt et al., 2018), direct use in Hierarchical RL (HRL) where a high-level controller is tasked to combine primitive skills or options (Sutton et al., 1999; Heess et al., 2016; Bacon et al., 2017; Wulfmeier et al., 2019; Daniel et al., 2012; Peng et al., 2019), transfer via the training objective such as regularisation towards expert behavior (Ross et al., 2011; Galashov et al., 2019; Tirumala et al., 2020) and transfer via the data generated by executing skills (Riedmiller et al., 2018; Campos et al., 2021; Torrey et al., 2007).

**Fine-tuning** often underperforms because neural network policies often do not easily move away from previously learned solutions (Ash & Adams, 2020; Igl et al., 2020; Nikishin et al., 2022) and hence may not easily adapt to new settings. As a result some work has focused on using previous solutions to 'kickstart' learning and improve on sub-optimal experts (Schmitt et al., 2018; Jeong et al., 2020; Abdolmaleki et al., 2021). When given multiple skills, fine-tuning can be achieved by reloading of parameters via a mixture (Daniel et al., 2012; Wulfmeier et al., 2019) or product (Peng et al., 2019) policy potentially with extra components to be learnt.

An alternative family of approaches uses the skill as a 'behavior prior' to to generate **auxiliary objectives** to regularize learning (Liu et al., 2021). This family of approaches have widely and successfully been applied in the offline or batch-RL setting to constrain learning (Jaques et al., 2019; Wu et al., 2019; Siegel et al., 2020; Wang et al., 2020; Peng et al., 2020). When used for transfer learning though, the prior can often be too constraining and lead to sub-optimal solutions (Rana et al., 2021).

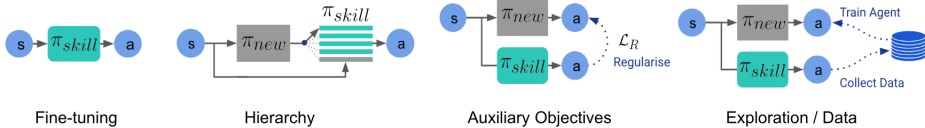

Figure 1: Transfer mechanisms for skills: Direct use via 1. fine-tuning or 2. hierarchical policies, 3. auxiliary training objectives, 4. exploration and data generation. Grey and green respectively denote new policy components and transferred skills.

Arguably the most widely considered use case for the reuse of skills falls under the umbrella of **hierarchical** RL. These approaches consider a policy that consists of two or more hierarchical 'levels' where a higher level controllers modulates a number of lower level skills via a latent variable or goal. This latent space can be continuous (Heess et al., 2016; Merel et al., 2019; Hausman et al., 2018; Haarnoja et al., 2018; Lynch et al., 2019; Tirumala et al., 2019; Pertsch et al., 2020; Ajay et al., 2021; Singh et al., 2021; Bohez et al., 2022), discrete (Florensa et al., 2017; Wulfmeier et al., 2020; Seyde et al., 2022) or a combination of the two (Rao et al., 2022).

A major motivation of such hierarchical approaches is the ability for different levels to operate at different timescales which, it is argued, could improve exploration and ease credit assignment. The most general application of this, is the Options framework (Sutton et al., 1999; Bacon et al., 2017; Wulfmeier et al., 2021) that operates using temporally-extended options. This allows skills to operate at longer timescales that may be fixed (Li et al., 2019; Zhang et al., 2020; Ajay et al., 2021) or with different durations (Bacon et al., 2017; Wulfmeier et al., 2021; Salter et al., 2022). The advantage of modeling such temporal correlations has been attributed to benefits of more effective exploration (Nachum et al., 2019). However, constraining the policy in such a way can often be detrimental to final performance (Bowling & Veloso, 1998; Jong et al., 2008).

Compared to other hierarchical methods such as the options framework, our method uses a final unconstrained, non-hierarchical solution and learn temporal abstraction as a task-specific instead of skill-specific property. In this way, we benefit from temporally correlated exploration without compromising on final asymptotic performance by distilling the final solution. Our work is closely related to **transfer via data** for off-policy multi-task RL as explored by (Riedmiller et al., 2018; Hafner et al., 2020) and initial work in transfer learning (Campos et al., 2021; Torrey et al., 2007). We extend these ideas to the compositional transfer of skills.

## 3    BACKGROUND

We frame our discussion in terms of the reinforcement learning problem in which an agent observes the environment, takes an action, and receives a reward in response. This can be formalized as a Markov Decision Process (MDP) consisting of the state space $\mathcal{S}$, the action space $\mathcal{A}$, and the transition probability $p(s_{t+1}|s_t, a_t)$ of reaching state $s_{t+1}$ from state $s_t$ when executing action $a_t$. The agent's behavior is given in terms of the policy $\pi(a_t|s_t)$. In the following, the agent's policy $\pi$ can either be parametrised by a single neural network or further be composed of $N$ skills $\pi_i(a|s)$ such that $\pi(a|s) = f(a|\{\pi_i\}_{i=0..N-1}, s)$.

The agent aims to maximize the sum of discounted future rewards, denoted by:

$$J(\pi) = \mathbb{E}_{\rho_0(s_0), p(s_{t+1}|s_t, a_t), \pi(a_t|s_t)}\Big[\sum_{t=0}^{\infty} \gamma^t r_t\Big], \tag{1}$$

where $\gamma$ is the discount factor, $r_t = r(s_t, a_t)$ is the reward and $\rho_0(s_0)$ is the initial state distribution. Given a policy we can define the state-action value function $Q(s_t, a_t)$ as the expected discounted return when taking an action $a_t$ in state $s_t$ and then following the policy $\pi$ as:

$$Q(s_t, a_t) = r(s_t, a_t) + \gamma \mathbb{E}_{p(s_{t+1}|s_t, a_t), \pi(a|s_t)}[Q(s_{t+1}, a)]. \tag{2}$$

## 4    METHOD

The method we propose in this paper, named *Skill Scheduler (SkillS)* (Fig. 2a) explicitly separates the data collection and task solution inference processes in order to better accommodate the require-

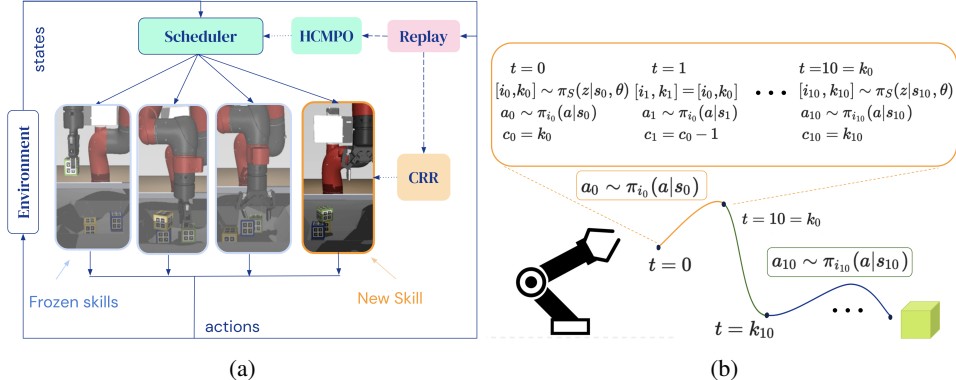

(a)                                                              (b)

Figure 2: **a) The Skill Scheduler framework.** Data is collected by the scheduler by sequencing selected skills among a set of pre-trained (frozen) skills and the new skill. This data is used to improve the scheduler using HCMPO (see Section 4.3) and the new skill via CRR. **b) Scheduler action and counter sampling.** At time $t = 0$ the scheduler samples an action $z_0 = [i_0; k_0]$ i.e. the skill with index $i_0$ is applied to the environment for $k_0$ time steps. When $(t = k_0)$, the scheduler samples a new action, the counter is set to the new length $c_t = k_t$ and decreased every time step.

ments of each. The data collection is performed by the *scheduler*, a high-level policy that chooses the best skills to maximize the task reward obtained from the environment. The task solution is learned via an off-policy learning algorithm that optimises a *new skill* $\pi_N(a|s, \phi)$ for the given task.

By avoiding assumptions about the representation and source of the underlying skills, we can use these existing behaviors agnostic to how they were generated. The agent has no access to any prior information on the quality of the skills and their relevance for the new task. While most of our experiments in Section 5 focus on reusing skills that are useful for the task, we conduct a detailed ablation study that investigates the robustness with respect to the number and quality of skills used.

### 4.1 SCHEDULER

The scheduler is represented by a multivariate categorical policy $\pi_S(z = [i; k]|s, \theta)$ that outputs an action $z$ consisting of:

- the index $i \in [0, .., N]$ of the skill to execute among the available skills $\bar{S} = S \cup \{\pi_N(a|s, \phi)\}$, where $S = \{\pi_i(a|s)\}_{i=0}^{N-1}$ are the $N$ previously trained and frozen skills and $\pi_N(a|s, \phi)$ is the new skill, and

- the number of steps the skill is executed for $k \in K$ (the *skill length*), where $K$ is a set of available skill lengths (positive integers) with cardinality $|K| = M$.

Whenever a new action $z = [i; k]$ is chosen by the scheduler, the skill $\pi_i(a|s)$ is executed in the environment for $k$ timesteps. The scheduler does not sample any new action until the current skill length $k$ is expired (see Fig. 2b). Choosing $k > 1$ means that the scheduler selects a new skill at a coarser timescale than the environment timestep. This induces temporal correlations in the actions, which are important for successful skill-based exploration in sparse-reward tasks (see Section 5).

Choosing one skill for multiple timesteps affects the agent's future action distributions and therefore future states and rewards. We add an auxiliary *counter* variable $c$ to the observations to retain the Markov property. Whenever the scheduler chooses a new action $z = [i; k]$ the counter is set equal to the chosen skill duration $c = k$. It is then decreased at every time step ($c_{t+1} = c_t - 1$) until the chosen skill duration has been reached ($c = 0$) and the scheduler chooses a new action (see Fig. 2b). The counter is crucial for learning of the critic, as detailed in Section 4.3.

Data is collected in the form $(s_t, a_t, z_t, c_t, r_t, s_{t+1})$ and is used both to train the scheduler and the new skill. The scheduler is trained only using the high-level actions $(\mathbf{z_t}, c_t, r_t, s_{t+1})$ with the accumulated task return over the episode as objective (Eq. 1).

We introduce an off-policy algorithm, *Hierarchical Categorical MPO* (HCMPO) to train the scheduler. While any RL algorithm could be used as a starting point, HCMPO builds upon a variant of Maximum A-Posteriori Policy Optimisation (MPO) (Abdolmaleki et al., 2018) with a discrete action space (Neunert et al., 2020). More details are provided in Section 4.3.

## 4.2 NEW SKILL

The new skill $\pi_N(a|s, \phi)$ interacts with the environment only when chosen by the scheduler. It is trained with the data only including the low-level action $(s_t, \mathbf{a_t}, r_t, s_{t+1})$ collected by the frozen skills S and the new skill itself according to the scheduler decisions. Given the strongly off-policy nature of this data[1], offline RL algorithms are most suited for learning the new skill. For this work, we use Critic Regularized Regression (Wang et al., 2020).[2]

Note that, although the scheduler and the new skill are trained with the same objective of maximizing the task reward (Eq. 1), we consider the new skill to provide the final solution for the task at hand, as it can reach higher performance. Whereas the scheduler acts in the environment via prelearned skills *fixed* to avoid degradation of the prior behaviors, the new skill is *unconstrained* and can thus fully adapt to the task at hand. In Section 5.2.3 we show how the scheduler and the new skill successfully tackle the two different problems of rapidly collecting task-relevant data (the scheduler) and slowly distilling the optimal task solution from the gathered experience (the new skill).

## 4.3 TRAINING THE SCHEDULER

In this section we derive the policy evaluation and improvement update rules of our proposed algorithm, Hierarchical Categorical MPO (HCMPO), used for training the scheduler.

**Policy Evaluation** In order to derive the updates, we take into account the fact that if $k_t$ time steps have passed, i.e. the counter $c_t$ has expired, the scheduler chooses a new action $z_t = [i_t; k_t]$ at time $t$. It keeps the previous action $z_t = z_{t-1} = [i_{t-1}; k_{t-1}]$ otherwise.

$$Q(s_t, c_t, i_t, k_t) \leftarrow r_t + \gamma \begin{cases} E_{\pi_S(i;k|s_{t+1})}[Q(s_{t+1}, c_{t+1}, i, k, \theta)] \text{ if } c_{t+1} = 1 \\ Q(s_{t+1}, c_{t+1} = c_t - 1, i_{t+1} = i_t, k_{t+1} = k_t) \text{ otherwise.} \end{cases} \quad (3)$$

If the scheduler has chosen a new action in the state $s_{t+1}$, the critic in state $s_t$ is updated with the standard MPO policy evaluation, i.e. with the critic evaluated using actions sampled from the current scheduler policy in the next state $i, k \sim \pi_{S,\theta}(i; k|s_{t+1})$. In all the other states instead, the critic is updated with the critic computed using the action $z_{t+1}$ actually taken by the scheduler, which is $z_{t+1} = z_t = [i_t; k_t]$. More details on the derivations of Eq. 3 are available in Appendix A.1.

**Policy Improvement** The scheduler policy is improved via the following mechanism but only on transitions where a new action is sampled by the scheduler:

$$\pi_S^{n+1} = \arg\max_{\pi_S} \int \int q(z|s) \log(\pi_S(z|s, \theta)dzds \text{ with } q(z|s) \propto \pi_S^n(z|s) \exp\left(Q^{\pi_S^n}(s, \mathbf{c} = \mathbf{1}, z)\right),$$
$$(4)$$

where $n$ is the index for the learning updates.

## 4.4 DATA AUGMENTATION

Eq. 3 and 4 show that only a subset of the available data would be actively used for learning the scheduler, i.e. only those states $s$ where the scheduler chooses a new action $z = [i; k]$. This effect is even stronger if the skill lengths $k \in K$ are $>> 1$, as a new scheduler action $z$ is sampled less frequently. While large skill lengths induce temporal correlations that are important for skill-based exploration, it has the side effect of decreasing the data actively contributing to improve the scheduler. In order to maximize sample efficiency without sacrificing exploration, we duplicate trajectories with long skill executions by also considering the same skill being executed repetitively

---

[1]The new skill is initially rarely executed as it commonly provides lower rewards than reloaded skills.
[2]We present an analysis of the advantages of using offline methods like CRR against off-policy methods like MPO in Appendix D.1.

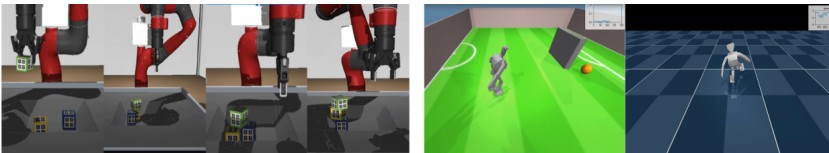

Figure 3: **Domains:** The experimental domains used: (from left to right) manipulation - Lift, Stack, Pyramid, Triple Stack and locomotion - GoalScoring and GetUpAndWalk.

for shorter amount of time[3]. More details regarding the augmentation are described in Appendix A.2 and we include ablations in Appendix D.2.

## 5 EXPERIMENTS

**Environments**   We empirically evaluate our method in robotic manipulation and locomotion domains. The goal of our evaluation is to study how different skill transfer mechanisms fare in new tasks which require skill reuse. In the manipulation setting (Fig. 3 (left)), we utilise a simulated Sawyer robot arm equipped with Robotiq gripper and a set of three objects - green (g), yellow (y) and blue (b) (Wulfmeier et al., 2020). We consider four tasks of increasing complexity: Lift g, Stack g on y, building a Pyramid of g on y and b, and Triple stack with g on y and y on b. Every harder task leverages previous task solutions as skills: this is a natural transfer setting to test an agent's ability to build on previous knowledge and solve increasingly complex tasks. For locomotion (Fig. 3 (right)), we consider two tasks with a simulated version of the OP3 humanoid robot (Robotis OP3): GetUpAndWalk and GoalScoring. The GetUpAndWalk task requires the robot to compose two skills: one to get up off of the floor and one to walk. In the GoalScoring task, the robot gets a sparse reward for scoring a goal, with a wall as an obstacle. The GoalScoring task uses a single skill but is transferred to a setting with different environment dynamics. This allows us to extend the study beyond skill composition to skill adaptability; both of which are important requirements when operating in the real world. All the considered transfer tasks use sparse rewards[4] except the GetUpAndWalk task where a dense walking reward is given but only if the robot is standing. We consider an off-policy distributed learning setup with a single actor and experience replay. For each setting we plot the mean performance averaged across 20 seeds with the shaded region representing one standard deviation. More details on the skills and tasks can be found in Appendix C.

**Analysis details**   Our method SkillS consists of two learning processes, one designed to quickly collect useful data (the scheduler) and one to distill the best solution for the task from the same data (the new skill). As our primary interest is in the task solution, unless otherwise specified, we focus on the *performance of the new skill* when executed in the environment in isolation for the entire episode. See Appendix C.2 for details about how the new skill and the scheduler can be separately evaluated. We compare SkillS against the following skill reuse methods: Fine-tuning, RHPO, KL-regularisation (KL-reg. to Mixture) and DAGGER. Hierarchical RL via RHPO (Wulfmeier et al., 2020) trains Gaussian mixture policies, and we adapt this method to reload skills as the components of the mixture. We can then either freeze the skills and include a new learned component (RHPO) or continue training the reloaded skills (fine-tuning). KL-reg. to Mixture adapts the MPO algorithm (Abdolmaleki et al., 2018) to include a regularisation objective to distill the reloaded skills; and DAGGER adapts Ross et al. (2011) for imitation learning with previous skills. A full description of our baseline methods can be found in Appendix B.

### 5.1 ANALYSIS: SPARSE REWARD TASKS IN LOCOMOTION AND MANIPULATION

Fig. 4 shows the performance of skill-based approaches on manipulation domains[5]. We observe that among all the considered methods, only SkillS can learn effectively in these domains, even

---

[3]For example, a trajectory of a skill executed for 100 timesteps could be generated equivalently by executing it once for 100 steps or 10 times consecutively for 10 steps each.

[4]When the reward provides a reliable learning signal (e.g. in case of dense rewards) there is less need to generate good exploration data using previous skills. The advantage of skill-based methods is thus primarily in domains where defining a dense, easier-to-optimise reward is challenging. For more details see Appendix D.3.

[5]Given the sparse nature of the reward, most methods fail to learn at all and appear as flat lines in the plot.

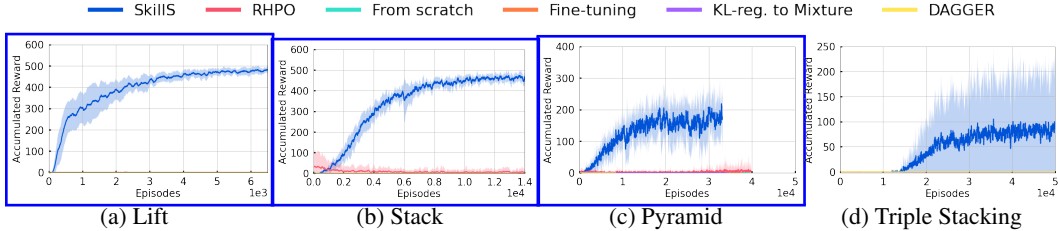

Figure 4: Performance of various skill transfer mechanisms on sparse-reward manipulation tasks. For each task, all approaches are given the previously trained useful skills (see Appendix for details) to learn the new task. SkillS is the only method to consistently learn across all tasks.

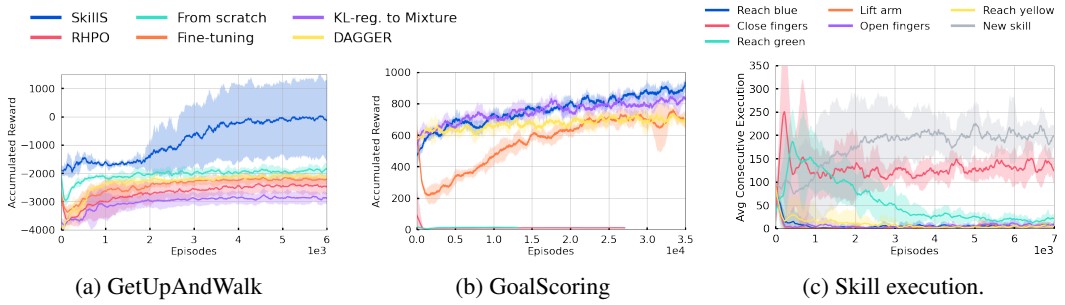

Figure 5: (a) - (b) Performance on skill transfer methods on locomotion tasks. SkillS outperforms other methods in GetUpAndWalk (a) and is no worse than other methods in the single skill GoalScoring task (b). (c) Skill average execution (as number of timesteps) during training.

though other methods have access to the same set of skills. When given access to a Lift task, all baselines still struggle to learn how to Stack. In contrast, SkillS can even learn challenging tasks like TripleStack with enough data. Importantly this illustrates that skill reuse on its own is insufficient to guarantee effective transfer, and highlights the strong benefit of temporal abstraction. While other methods such as RHPO switch between skills at high frequency, the scheduler's action space encourages the execution of skills over longer horizons, considerably reducing the search space.

Fig. 5a-b compares performance in the locomotion setting. In the GetUpAndWalk task, we find SkillS outperforms the other methods, confirming that temporal abstraction is fundamental when solving tasks that require a sequence of skills to be learned (getting up and then walking in this task). Executing the skills for longer horizons also generates higher reward early during training. Particularly poor is the performance of regularising towards a mixture of these skills as it performs worse than learning from scratch. For the GoalScoring task which requires transferring a single skill to new dynamics, SkillS is competitive and shares the strongest results. As we will demonstrate in Section 5.2.4, the advantage of temporally correlated exploration conferred by SkillS is less important in settings with only one skill.

## 5.2 ABLATIONS

In this section we analyse the robustness of SkillS and evaluate the importance of various components and design decisions. In particular, we analyse (1) the scheduler skill selection during training; (2) the robustness to the number and quality of skills used; (3) the benefit and flexibility of having a separate infer mechanism and (4) the utility of flexible temporal abstraction. More ablations including a study of dense and sparse rewards, the benefit of offline methods like CRR and the importance of data augmentation can be found in Appendix D.

### 5.2.1 SCHEDULER SKILL SELECTION

In this section we show how the scheduler typically chooses the skills to interact with the environment. Fig. 5c displays the skill selection during the training by means of the average *consecutive* execution for each skill. We assume 7 skills are available, some of them needed for solving the task

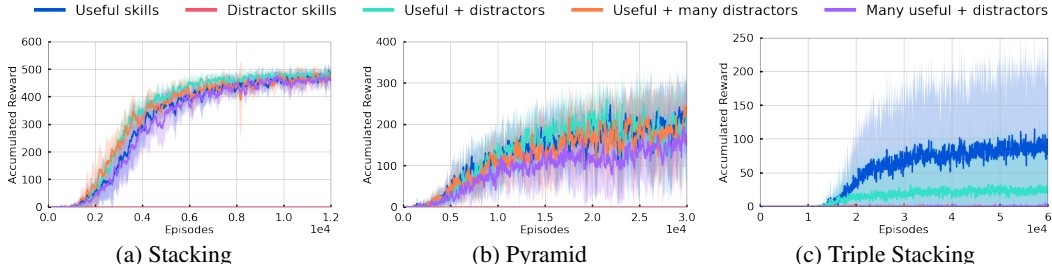

Figure 6: SkillS can learn even when the number and type of skills change although performance deteriorates on harder tasks.

at hand ('reach green', 'close fingers'), while the others act as a distractor (e.g. 'reach yellow' or 'open fingers'). At the beginning the scheduler quickly selects the most useful skills ('reach green', 'close fingers') while discarding those less relevant to the task (observe the drop in executing 'reach yellow' or 'reach blue'). While the scheduler does not choose the poorly performing new skill early in training, eventually (after roughly 2000 episodes) it is selected more often until finally it is the most executed skill.

### 5.2.2 ROBUSTNESS TO SKILL NUMBER AND QUALITY

Fig. 6 evaluates SkillS on the stacking, pyramid and triple-stacking domains while changing the skill set $S$. For this analysis we vary the number[6] and the types of skills by including a variety of *distractor* skills not directly relevant for the task (e.g. reach 'blue' when 'red' needs to be stacked; more details in Appendix C.1.3). With only distractor skills, SkillS understandably fails to learn anything. For the easier tasks of Stacking and Pyramid, SkillS is quite robust as long as there are *some useful* skills in the set $S$. On the challenging triple-stacking task, increasing the number of skills has a detrimental effect on performance. We hypothesize that the larger skill set increases the size of the search space to a point where the exploration in skill space does not easily find the task solution.

### 5.2.3 BENEFIT OF SEPARATE COLLECT AND INFER MECHANISMS

We conduct an experiment to study the importance of a separate *inference* (learning) mechanism in extension of the hierarchical agent - the scheduler. Fig. 7a compares the performance of the two components of SkillS: 'Scheduler (incl. new skill)' which has access to the new skill and frozen skills and 'New skill' which is the inferred new skill. In addition, we include a version 'Scheduler (without new skill)' where the higher level scheduler *only* has access to the frozen skills. As the figure shows, having a separate off-policy learning process is crucial for learning as the new skill (blue) outperforms the scheduler in performance, even when it can execute the new skill itself (red). The gap in performance is even more evident if the scheduler cannot access the new skill (light blue).

### 5.2.4 UTILITY OF FLEXIBLE TEMPORAL ABSTRACTION

**Temporally-correlated exploration** Figs. 7b and 7c illustrate the importance of temporally-correlated exploration that arises from our formulation in Section 4. For this analysis we additionally run a version of our algorithm called 'SkillS (without temporal abstraction)' where a new skill must be chosen at each timestep ($K = \{1\}$). When given a single (near-)optimal skill[7], all methods including regularisation and fine-tuning perform similarly (Fig. 7b). However, when we use more than one sub-optimal skill, temporally correlated exploration is crucial for learning (Fig. 7c).

**Variable temporal abstraction** Fig. 8 analyses the importance of the flexible temporal abstraction of SkillS by comparing our method against a version dubbed 'SkillS (fixed temporal abstraction - X)' where each skill, once chosen, is held fixed for a given duration of X timesteps ([10, 50, 100,

---

[6]In order to augment the number of skills, we use 3 skills trained to solve the same task (e.g. reach 'blue') from different seeds and we refer to these sets in Fig. 6 as *many distractors* and *many useful skills*.

[7]The near-optimal skill for the Stacking task places the green object on the yellow one without moving the gripper away.

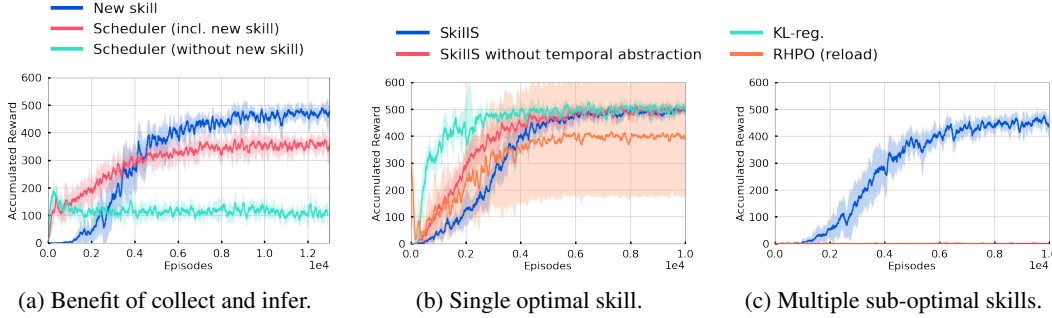

(a) Benefit of collect and infer.    (b) Single optimal skill.    (c) Multiple sub-optimal skills.

Figure 7: (a) SkillS performs best when given access to a separately inferred new skill. (b)-(c) The main advantage of SkillS is likely due to temporally correlated exploration. When given access to a single optimal skill (b) all methods can learn effectively but with multiple sub-optimal skills (c), SkillS is the only method to learn (all other methods are flat lines). All results are on the Stacking task.

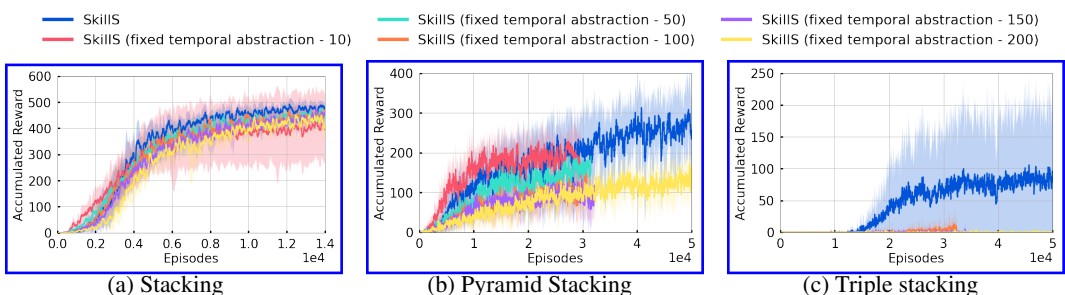

(a) Stacking    (b) Pyramid Stacking    (c) Triple stacking

Figure 8: The variable temporal abstraction of SkillS is especially crucial for learning harder tasks.

150, 200] timesteps in a 600 step episode). We observe that the optimal fixed skill length varies from task to task (see Fig. 8a vs 8b) and, most importantly, the performance on harder tasks (especially triple-stacking) degrades when using this constraint. Thus, flexibly choosing the duration for each skill is quite important.

# 6    DISCUSSION

Transferring the knowledge of reinforcement learning agents via skills or policies has been a long-standing challenge. Here, we investigate different types of task and dynamics variations and introduce a new method, SkillS, that learns using data generated by sequencing skills. This approach demonstrates strong performance in challenging simulated robot manipulation and locomotion tasks. It further overcomes limitations of fine-tuning, imitation and purely hierarchical approaches. It achieves this by explicitly using two learning processes. The high-level scheduler can learn to sequence temporally-extended skills and bias exploration towards useful regions of trajectory space, and the new skill can then learn an optimal task solution off-policy.

There are a number of interesting directions for future work. For instance, unlike in the options framework (Sutton et al., 1999) which associates a termination condition with each option, in our work the duration of execution of a skill is independent of the pretrained skill. While this provides more flexibility during transfer, it also ignores prior knowledge of the domain, which may be preferable in some settings. Similarly, the high-level controller is currently trained to maximize task reward. However, it may be desirable to optimize the scheduler with respect to an objective that incentivizes exploration, e.g. to reduce uncertainty with regards to the value or system dynamics, improving its ability to collect data in harder exploration problems. Finally, our approach can naturally be combined with transfer of forms of prior knowledge apart from skills, such as prior experience data, the value function (Galashov et al., 2020), or a dynamics model (Byravan et al., 2021) and we expect this to provide fertile area of future research.

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

APPENDIX

## A  SKILLS DETAILS

### A.1  HCMPO - POLICY EVALUATION

In order to derive the PE updates we write down the probability that the scheduler chooses an action $z_t$ and the counter $c_t$.

This takes into account the fact that, if $k_t$ time steps have passed, i.e. the counter $c_t$ has expired, the scheduler chooses a new action $z_t = [i_t; k_t]$ at time $t$. It keeps the previous action $z_t = z_{t-1} = [i_{t-1}; k_{t-1}]$ otherwise.

$$\pi(z_t, c_t | s_t, z_{t-1}, c_{t-1}) =$$

$$\pi(i_t, k_t, c_t | s_t, i_{t-1}, k_{t-1}, c_{t-1}) = \begin{cases} \pi_S(z_t = [i_t; k_t] | s_t, \theta) \delta(k_t, c_t) \text{ if } c_{t-1} = 1 \\ \delta(i_t - i_{t-1}) \delta(k_t - k_{t-1}) \delta(c_t - (c_{t-1} - 1)) \text{ otherwise.} \end{cases}$$

(5)

We can now use Eq. 5 to derive the PE update rules reported in Eq. 3.

### A.2  DATA AUGMENTATION

We define a *skill trajectory* $\tau$ as the sequence of $k^*$ time steps where the same skill with index $i^*$ is executed

$$\tau = \{(s_t, z_t = [i^*, k^*], c_t, r_t)\}_{t=t_0}^{t=t_0+k^*-1}$$

(6)

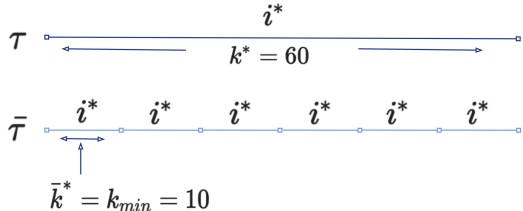

Figure 9: **Example of data augmentation.** Let's assume we have a trajectory $\tau$ where the skill with index $i^*$ is executed for $k^* = 60$ time steps. The duplicated trajectory $\bar{\tau}$ is generated by pretending the scheduler chose the same skill with index $i^*$ every $k_{min}$ time steps, equal to 10 in our example. This generates a new trajectory $\bar{\tau}$ compatible with the low-level actions $a_t$ and state $s_t$ actually collected during the interaction with the environment.

and $k_{min}$ the smallest skill length available to the scheduler ($k_{min} \leq k$ for $k \in \mathrm{K}$ and $k_{min} \in \mathrm{K}$ ).

For every trajectory $\tau$ with skill length $k^*$ that can be written as multiple of the minimum skill length $k_{min}$ ($mod(k_t, k_{min}) = 0$), we generate a new *duplicated skill trajectory* $\bar{\tau}$ pretending it was generated by a scheduler choosing the same skill $i^*$ multiple times with the smallest available skill length $k_{min}$ (see Fig. 9). The *duplicated skill trajectory* $\bar{\tau}$

$$\bar{\tau} = \{(s_t, \bar{z}_t = [i^*, \bar{k}^*], \bar{c}_t, r_t)\}_{t=t_0}^{t=t_0+k^*-1} \tag{7}$$

is such that:

$$(s_t, \bar{z}_t = [i^*, \bar{k}^*], \bar{c}_t, r_t) =$$

$$\begin{cases} (s_t, \bar{z}_t = [i^*, k_{min}], c_t - \lfloor(\frac{c_t}{k_{min}} - 1) * k_{min}, r_t) \text{ if } mod(k_t, k_{min}) = 0, c_t > k_{min} \\ (s_t, \bar{z}_t = [i^*, k_{min}], c_t, r_t) \text{ if } mod(k_t, k_{min}) = 0, c_t \leq k_{min} \\ (s_t, z_t = [i^*, k^*], c_t, r_t) \text{ otherwise.} \end{cases} \tag{8}$$

This generates a new trajectory that is compatible with the data actually collected, but with more samples that are actively used for improving the scheduler and the policy, as it pretends the scheduler made a decision in more states. Both $\tau$ and $\bar{\tau}$ are added to the replay and used for training the scheduler. This perspective is highly related to efficiently learning high-level controllers over sets of options in the context of intra-option learning (using data throughout option execution to train the inter option policy) (Sutton et al., 1999; Wulfmeier et al., 2021).

### A.3 Bias on initial skill length

Our SkillS formulation enables the possibility of specifying initial bias in the scheduler choices. It is possible to encourage the scheduler to *initially* choose one or more skills and/or skill lengths. In all our experiments we bias the scheduler to initially choose the largest skill length. This way all the skills are executed for a considerably large number of timesteps at the beginning of training, helping the scheduler understand their utility for the task at hand.

## B Baseline details

In this section we describe the methods used as baselines in the main text. For all of the baselines described below, we use a weighted loss that trades off the baseline with CRR using a parameter $\alpha$. In other words when $\alpha = 1$, we use pure CRR whereas $\alpha = 0$ runs the pure algorithm.

### B.1 MPO

When learning from scratch and for our baselines we largely adapt the state-of-the-art off-policy learning algorithm Maximum a-posteriori Policy Optimisation (MPO) (Abdolmaleki et al., 2018). MPO optimises the RL objective in Eq. 1 using an Expectation Maximisation (EM) style approach. The algorithm operates in three steps: a policy evaluation step to learn the state-action value function

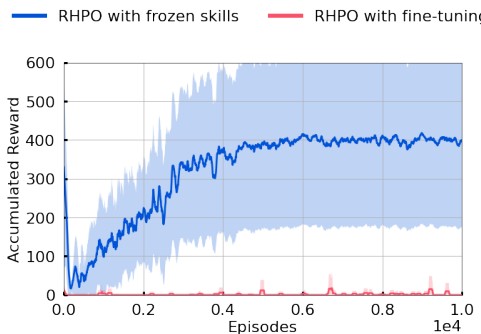

Figure 10: Impact of freezing skills: Single skill transfer to stacking.

$Q^{\pi}(s, a)$; an E-step to create an improved non-parametric policy using data sampled from a replay buffer; and an M-step that fits a parametric policy $\pi(a|s)$ to the non-parametric estimate.

For the policy evaluation step we use a Mixture of Gaussian (MoG) critic as described by Shahriari et al. (2022). MPO then optimizes the following objective in the E-step:

$$\max_q \int_s \mu(s) \int_a q(a|s)Q(s, a)\, da\, ds$$
$$s.t. \int_s \mu(s)D_{\mathrm{KL}}(q(a|s)||\pi(a|s))\, ds < \epsilon, \tag{9}$$

where $q(a|s)$ an improved non-parametric policy that is optimized for states $\mu(s)$ drawn from the replay buffer. The solution for $q$ is shown to be:

$$q(a|s) \propto \pi(a|s, \theta) \exp(\frac{Q_\theta(s, a)}{\eta^*}). \tag{10}$$

Finally in the M-step, an improved parametric policy $\pi^{n+1}(a|s)$ is obtained via supervised learning:

$$\pi^{n+1} = \arg\max_{\pi_\theta} \sum_j^M \sum_i^N q_{ij} \log \pi_\theta(a_i|s_j).$$
$$s.t. D_{\mathrm{KL}}(\pi^n(a|s_j)||\pi_\theta(a|s_j))$$

Importantly, the M-Step also incorporates a KL-constraint to the previous policy $\pi^n(a|s)$ to create a trust-region like update that prevents overfitting to the sampled batch of data. In our experiments, we use an extension of MPO, dubbed Regularized Hierarchical Policy Optimisation (RHPO), that learns a Mixture of Gaussian policy (Wulfmeier et al., 2020) instead of a single isotropic Gaussian.

### B.2 FINE-TUNING AND RHPO

For our comparison of skill reuse, we reload the mixture components of an RHPO agent using the parameters of the previously trained skills. In addition we consider two scenarios: a) continuing to fine-tune the mixture components that were reloaded or b) introducing a new mixture component (similar to the new policy of SkillS) and freezing the other mixture components that were reloaded using skills. Freezing the skills is often advantageous to prevent premature over-writing of the learnt behavior - an important feature which can add robustness early in training when the value function is to be learnt. We illustrate this effect in Figure 10 in the 'Stacking' setting where we transfer only a single skill. As shown, when the previous skill is fixed the retained knowledge can be used to bootstrap learning whereas direct fine-tuning can cause a catastrophic forgetting effect where learning performance is significantly worse.

### B.3 KL-REGULARISATION

KL-regularisation typically involves the addition of an extra term to the RL objective function that minimises the Kullback-Leibler (KL) divergence between the policy and some prior distribution

(Tirumala et al., 2020). Here, we define the prior distribution to be a uniform mixture over the previously trained skills, given by:

$$\pi_0 = \sum_{i=1}^{N} w_i \times \pi_i(a|s) \tag{11}$$

$$w_i = \frac{1}{N} \forall i$$

where $\pi_i(a|s)$ are $N$ previously trained skills.

As such, the KL objective can be incorporated into any RL algorithm. In this work we adapt MPO to include regularisation to a prior. As presented above, MPO includes KL-constraints in both the E and M steps, either of which can be used to further regularise to the prior. In this work, we adapt the E-step to regularise against a mixture distribution. In this setting, the non-parametric distribution $q$ for the E-step can be obtained as:

$$q(a|s) \propto \pi_0(a|s, \theta) \exp(\frac{Q_\theta(s, a)}{\eta^*}). \tag{12}$$

One interpretation of this objective is that the prior $\pi_0$ serves as a *proposal* distribution from which actions are sampled and then re-weighted according to the exponentiated value function. The optimal temperature $\eta^*$ here is obtained by minimising:

$$g(\eta) = \eta\epsilon$$
$$+ \eta \int \mu(s) \log \int \pi_0(a|s, \theta) \exp(\frac{Q_\theta(s, a)}{\eta}), \tag{13}$$

where $\epsilon$ is a hyper-parameter that controls the tightness of the KL-bound. A smaller $\epsilon$ more tightly constrains the policy to be close to the prior distribution whereas a looser bound allows more deviation from the prior and reliance on the exponentiated Q function for learning.

### B.4 DAGGER

The final baseline we consider is an adaptation of the method DAGGER (Dataset Aggregation) introduced by Ross et al. (2011). DAGGER is an imitation learning approach wherein a student policy is trained to imitate a teacher. DAGGER learns to imitate the teacher on the data distribution collected under the student. In our formulation, the teacher policy is set to be a uniform mixture over skills, $\pi_0$ as defined in Eq. 11. We train the student to minimize the cross-entropy loss to this teacher distribution for a state distribution generated under the student. The loss for DAGGER is thus given by:

$$L_{DAGGER} = \arg\min_{\pi} -\pi_0(a|s) \log \pi(a|s) \tag{14}$$

In addition, we consider an imitation + RL setting where the DAGGER loss is augmented with the standard RL objective using a weighting term $\alpha_{DAGGER}$:

$$L = \alpha_{DAGGER} \times L_{DAGGER} + (1 - \alpha_{DAGGER}) \times L_{MPO}. \tag{15}$$

In our experiments we conduct a hyper-parameter sweep over the parameter $\alpha_{DAGGER}$ and report the best results.

## C EXPERIMENT DETAILS

### C.1 TASKS AND REWARDS

#### C.1.1 MANIPULATION

In all manipulation tasks, we use a simulated version of the Sawyer robot arm developed by Rethink Robotics, equipped with a Robotiq 2F85 gripper. In front of the robot, there is a basket with a base size of 20x20 cm and lateral slopes. The robot can interact with three cubes with side of 4 cm, coloured respectively in yellow, green and blue. The robot is controlled in Cartesian velocity space

| Entry | Dimension | Unit |
|---|---|---|
| End-effector Cartesian linear velocity | 3 | m/s |
| Gripper rotational velocity | 1 | rad/s |
| Gripper finger velocity | 1 | rad/s |

Table 1: **Robot action space in manipulation tasks.**

| Entry | Dimension | Unit |
|---|---|---|
| Joint Angle | 7 | rad |
| Joint Velocity | 7 | rad/s |
| Pinch Pose | 7 | m, au |
| Finger angle | 1 | rad |
| Finger velocity | 1 | rad/s |
| Grasp | 1 | n/a |
| Yellow cube Pose | 7 | m, au |
| Yellow cube pose wrt Pinch | 7 | m, au |
| Green cube Pose | 7 | m, au |
| Green cube pose wrt Pinch | 7 | m, au |
| Blue cube Pose | 7 | m, au |
| Blue cube pose wrt Pinch | 7 | m, au |

Table 2: **Observations used in simulation.** In the table au stands for arbitrary units used for the quaternion representation.

with a maximum speed of 0.05 m/s. The arm control mode has four control inputs: three Cartesian linear velocities as well as a rotational velocity of the wrist around the vertical axis. Together with the gripper opening angle, the total size of the action space is five (Table 1). Observations used for our experiments are shown in Table 2. For simulating the real robot setup we use the physics simulator MuJoCo (Todorov et al., 2012). The simulation is run with a physics integration time step of 0.5 milliseconds, with control interval of 50 milliseconds (20 Hz) for the agent.

### C.1.2 LOCOMOTION

We use a Robotis OP3 (Robotis OP3) humanoid robot for all our locomotion experiments. This hobby grade platform is 510 mm tall and weighs 3.5 kg. The actuators operate in position control mode. The agent chooses the desired joint angles every 25 milliseconds (40 Hz) based on on-board sensors only. These observations include the joint angles, and the angular velocity and gravity direction of the torso. Gravity is estimated from IMU readings with a Madgwick (Madgwick et al., 2010) filter that runs at a higher rate. The robot has also a web camera attached to the torso, but we ignore it in our experiments. To account for the latency in the control loop, we stack the last five observations and actions together before feeding them to the agent. Further, to encourage smoother exploration, we apply an exponential filter with a strength of 0.8 to the actions before passing them to the actuator. We also assume we have access to the robot's global position and velocity via an external motion capture system. This privileged information is, however, only used to evaluate the reward function, and is not visible to the agent. Although we train our policies in simulation, the model has been carefully tuned to match the physical robot, allowing us to zero-shot transfer the policies directly from simulation to a robot (Byravan et al., 2023).

### C.1.3 SKILLS

The skills used for each task are shown in Table 3. All the plots in the paper are shown using only the skills under the column 'Useful skills'. The other sets of skills are used for the ablation in Fig. 6. In particular, when in Fig. 6 we mention 'Many Useful skills' or 'Many Distractor skills' it means that we use 3 skills trained for the same task, e.g. "reach green", but from different seeds (e.g. 'Many useful skills and distractor skills' for Pyramid is a total of 31 skills).

| Task | Useful skills | No. | Distractor skills | No. |
|---|---|---|---|---|
| Lift (green) | Reach green, Lift arm with closed fingers | 2 | Reach yellow, Reach blue, Open fingers, Lift arm with open fingers | 4 |
| Stack (green on yellow) | Useful skills for Lift green, Lift green, Hover green on yellow | 4 | Distractor skills for Lift green, Lift yellow, Lift blue | 6 |
| Pyramid (green on top) | Useful skills for Stack, Stack green on yellow, Lift yellow, Lift blue Reach yellow, Reach blue | 9 | Lift arm with open fingers, Open fingers, Stack yellow on blue | 3 |
| Triple stacking (green on yellow on blue) | Useful skills for Stack, Stack green on yellow, Stack yellow on blue, Lift yellow, Reach yellow, Hover yellow on blue, | 9 | Lift arm with open fingers, Open fingers, Reach blue, Lift blue | 4 |

Table 3: **Skills used for each manipulation task.** Every column with title 'No.' counts the number of skills in the left-hand side column, e.g. 'Lift green' has 2 Useful skills.

### C.1.4   REWARDS - MANIPULATION

The skills of Table 3 are obtained either by training them from scratch with staged rewards (in particular, the basic skills such as 'Open fingers') or they are the result of a SkillS experiment, i.e. they are represented by the New Skill and have been trained with sparse rewards. The reward functions used for this aim are the following.

- **Reach object - dense**

$$R_{reach} = \begin{cases} 1 & \text{iff } ||\frac{p_{gripper}-p_{object}}{tol_{pos}}|| \leq 1 \\ 1 - \tanh^2(||\frac{p_{gripper}-p_{object}}{tol_{pos}}|| \cdot \epsilon) & \text{otherwise,} \end{cases} \quad (16)$$

  where $p_{gripper} \in \mathrm{R}^3$ is the position of the gripper, $p_{object} \in \mathrm{R}^3$ the position of the object, $tol_{pos} = [0.055, 0.055, 0.02]m$ the tolerance in position and $\epsilon$ a scaling factor.

- **Open fingers - dense**

$$R_{open} = \begin{cases} 1 & \text{iff } ||\frac{p_{finger}-p_{desired}}{tol_{pos}}|| \leq 1 \\ 1 - \tanh^2(||\frac{p_{finger}-p_{desired}}{tol_{pos}}|| \cdot \epsilon) & \text{otherwise,} \end{cases} \quad (17)$$

  where $p_{finger} \in \mathrm{R}$ is the angle of aperture of the gripper, $p_{desired} \in \mathrm{R}$ is the angle corresponding to the open position, $tol_{pos} = 1e^{-9}$ the tolerance in position and $\epsilon$ a scaling factor.

- **Lift arm with open fingers - dense**

$$R_{lift} = \begin{cases} 1 & \text{iff } z_{arm} \geq z_{desired} \\ \frac{z_{arm}-z_{min}}{z_{desired}-z_{min}} & \text{otherwise,} \end{cases} \quad (18)$$

  where $z_{arm} \in \mathrm{R}$ is the z-coordinate of the position of the gripper, $z_{min} = 0.08 \in \mathrm{R}$ is the minimum height the arm should lift to receive non-zero reward and $z_{desired} = 0.18 \in \mathrm{R}$ is the desired z-coordinate the position of the gripper should reach to get maximum reward. The reward to have both the arm moving up and the fingers open is given by:

$$R_{lift,open} = R_{lift} \cdot R_{open}. \quad (19)$$

- **Lift arm with closed fingers - dense**

$$R_{grasp} = 0.5(R_{closed} + R_{grasp,aux}), \quad (20)$$

  where $R_{closed}$ is computed with the formula of Eq. equation 17, but using as $p_{desired} \in \mathrm{R}$ the angle corresponding to *closed* fingers this time;

$$R_{grasp,aux} = \begin{cases} 1 & \text{if grasp detected by the grasp sensor} \\ 0 & \text{otherwise.} \end{cases} \quad (21)$$

Then the final reward is:

$$R_{lift,closed} = R_{lift} \cdot R_{grasp}. \tag{22}$$

- **Lift object - sparse**

$$R_{lift} = \begin{cases} 1 & \text{iff } z_{object} \geq z_{desired} \\ 0 & \text{otherwise,} \end{cases} \tag{23}$$

where $z_{object} \in \mathrm{R}$ is the z-coordinate of the position of the object to lift and $z_{desired} = 0.18$ the desired z-coordinate the object should reach to get maximum reward.

- **Hover object on another one - dense**

$$R_{hover} = \begin{cases} 1 & \text{iff } ||\frac{p_{top}-p_{bottom}-offset}{tol_{pos}}|| \leq 1 \\ 1 - \tanh^2(||\frac{p_{top}-p_{bottom}-offset}{tol_{pos}}|| \cdot \epsilon) & \text{otherwise} \end{cases}, \tag{24}$$

where $p_{top} \in \mathrm{R}^3$ is the position of the top object, $p_{bottom} \in \mathrm{R}^3$ the position of the bottom object, $offset = [0, 0, object_{height} = 0.04]$, $tol_{pos} = [0.055, 0.055, 0.02]m$ the tolerance in position and $\epsilon$ a scaling factor.

- **Stack top object on bottom object - sparse**

$$R_{stack} = \begin{cases} 1 & \text{iff } d_{x,y}(p_{top}, p_{bottom}) \leq tol_{x,y} \text{ \&} \\ & |d_z(p_{top}, p_{bottom}) - d_{desired}| \leq tol_z \text{ \&} \\ & R_{grasp,aux} = 0 \\ 0 & \text{otherwise,} \end{cases} \tag{25}$$

where $d_{x,y}(\cdot, \cdot) \in \mathrm{R}$ is the norm of the distance of 2D vectors (on $x$ and $y$), $d_z(\cdot, \cdot) \in \mathrm{R}$ is the (signed) distance along the z-coordinate, $p_{top} \in \mathrm{R}^3$ and $p_{bottom} \in \mathrm{R}^3$ are respectively the position of the top and bottom object, $tol_{x,y} = 0.03$ and $tol_z = 0.01$ are the tolerance used to check if the stacking is achieved, $d_{desired} = 0.04$ is obtained as the average of the side length of the top and bottom cubes.
The final reward we use for stacking is:

$$R_{stack,leave} = R_{stack} \cdot (1 - R_{reach,sparse}(top)) \tag{26}$$

where $R_{reach,sparse}(top)$ is the sparse version of $R_{reach}$ in Eq. equation 24 for the $top$ object.

- **Pyramid - sparse**

$$R_{pyramid} = \begin{cases} 1 & \text{iff } d_{x,y}(p_{bottom,a}, p_{bottom,b}) \leq tol_{x,y,bottom} \text{ \&} \\ & d_{x,y}(p_{top}, p_{bottom}) \leq tol_{x,y} \text{ \&} \\ & |d_z(p_{top}, p_{bottom}) - d_{desired}| \leq tol_z \text{ \&} \\ & R_{grasp,aux} = 0 \\ 0 & \text{otherwise,} \end{cases} \tag{27}$$

where $d_{x,y}(\cdot, \cdot) \in \mathrm{R}$ is the norm of the distance of 2D vectors (on $x$ and $y$), $d_z(\cdot, \cdot) \in \mathrm{R}$ is the (signed) distance along the z-coordinate, $p_{top}, p_{bottom,a}, p_{bottom,b} \in \mathrm{R}^3$ are respectively the position of the top object and the 2 bottom objects, while $p_{bottom} = \frac{p_{bottom,a}+p_{bottom,b}}{2} \in \mathrm{R}^3$ is the average position of the two bottom objects, $tol_{x,y,bottom} = 0.08$, $tol_{x,y} = 0.03$ and $tol_z = 0.01$ are the tolerance used to check if the pyramid configuration is achieved, $d_{desired} = 0.04$ is obtained as the average of the side length of the top and bottom cubes. The final reward we use for our tasks is:

$$R_{pyramid,leave} = R_{pyramid} \cdot (1 - R_{reach,sparse}(top)). \tag{28}$$

- **Triple stacking - sparse** The triple stacking reward is given by:

$$R_{triple,stack,leave} = R_{stack}(top, middle) \cdot R_{stack}(middle, bottom) \cdot (1 - R_{reach,sparse}(top)). \tag{29}$$

### C.1.5 Rewards - Locomotion

In the GetUpAndWalk task, we condition the robot's behavior on a goal observation: a 2-d unit target orientation vector, and a target speed in m/s. In these experiments, the speed was either 0.7m/s or 0m/s. The targets are randomly sampled during training. Each target is kept constant for an exponential amount of time, with an average of 5 seconds between resampling.

The reward per time step is a weighted sum $R_{walk}$ with terms as described below.

While this reward is dense in the observation space, 2/3 of episodes are initialized with the robot lying prone on the ground (either front or back). The effective sparsity results from the need to first complete the challenging get-up task, before the agent can start to earn positive reward from walking.

- **GetUpAndWalk**

$$R_{walk} = R_{orient} + R_{velocity} + R_{upright} + R_{action} + R_{pose} + R_{ground} + 1, \quad (30)$$

where $R_{orient}$ is the dot product between the target orientation vector and the robot's current heading; $R_{velocity}$ is the norm of the difference between the target velocity, and the observed planar velocity of the feet; $R_{upright}$ is 1.0 when the robot's orientation is close to vertical, decaying to zero outside a margin of $12.5°$; $R_{action}$ penalizes the squared angular velocity of the output action controls averaged over all 20 joints, $\sum_{joint} \left( \omega_t^{joint} - \omega_{t-1}^{joint} \right)^2$; $R_{pose}$ regularizes the observed joint angles toward a reference "standing" pose $\sum_{joint} \left( \theta^{joint} - \theta_{ref}^{joint} \right)^2$; and $R_{ground} = -2$ whenever the robot brings body parts other than the feet within 4cm of the ground.

In GoalScoring, the robot is placed in a $4m \times 4m$ walled arena with a soccer ball. It must remain in its own half of the arena. It scores goals when the ball enters a goal area, $0.5m \times 1m$, positioned against the centre of the back wall in the other half. An obstacle (a short stretch of wall) is placed randomly between the robot and the goal.

- **GoalScoring**

$$R_{goalscoring} = R_{score} + R_{upright} + R_{maxvelocity} \quad (31)$$

where: $R_{score}$ is 1000 on the single timestep where the ball enters the goal region (and then becomes unavailable until the ball has bounced back to the robot's own half); $R_{maxvelocity}$ is the norm of the planar velocity of the robot's feet in the robot's forward direction; $R_{upright}$ is the same as for GetUpAndWalk. Additionally, episodes are terminated with zero reward if the robot leaves its own half, or body parts other than the feet come within 4cm of the ground.

## C.2 Details on how we evaluate performance

Fig. 12 show how the performance of the scheduler and the new skill can be evaluated separately.

## C.3 Training and networks

For all our experiments we use Multilayer Perceptron (MLP) network torsos with a network head output that is twice the size of the action dimension for the task. The network output is split into the mean and log standard deviation that is used to parameterize the output of an isotropic Gaussian distribution. We use a single actor and learner, a replay buffer for off policy learning and a controller that ensures a fixed ratio between actor steps and learner updates across experiments. For each method and hyper-parameter setting results are averaged across 5 random seeds. In what follows we provide a detailed description of hyper-parameters used for the various experiments.

### C.3.1 Manipulation

The following parameters are fixed across all methods compared in the Manipulation setting (SkillS and baselines):

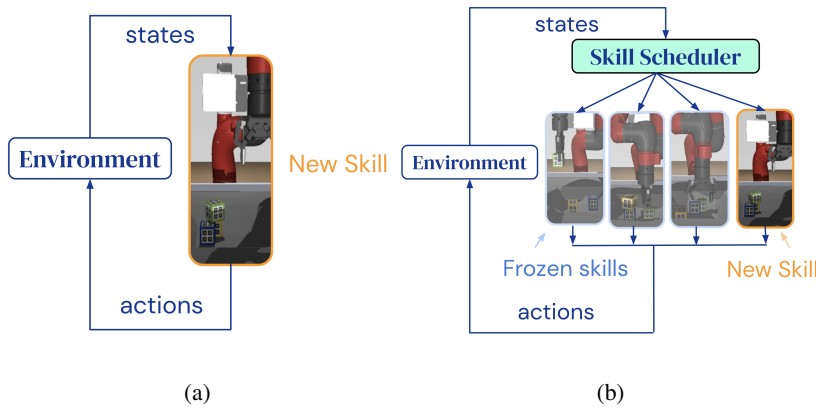

(a)  (b)

Figure 12: During evaluation, the new skill is executed in the environment for the entire episode (a). This differs from the way the data is collected at training time, where the new skill can interact with the environment only when selected by the scheduler (b).

**Learner parameters**

- Batch size: 256
- Trajectory length: 10
- Learning rate: 3e-4
- Min replay size to sample: 200
- Samples per insert: 50
- Replay size: 1e6
- Target Actor update period: 25
- Target Critic update period: 100
- E-step KL constraint $\epsilon$: 0.1
- M-step KL constraints: 5e-3 (mean) and 1e-5 (covariance)

**Network parameters**

- Torso for Gaussian policies: (128, 256, 128)
- Component network size for Mixture policies: (256, 128)
- Categorical network size for MoG policies: (128,)

**Scheduler specific parameters**

- Scheduler batch size: 128
- Scheduler learning rate: 1e-4
- Min replay size to sample for scheduler: 400
- Samples per insert: 100
- Scheduler target actor update period: 100
- Scheduler target critic update period: 25
- Available skill lengths: $K = \{n \cdot 10\}_{n=1}^{10}$
- Initial skill lengths biases: 0.95 for $k = 100$, 0.005 for the other skill lengths

**Hyper-parameter sweeps**

- $\alpha$ (MPO v/s CRR loss) - (0, 0.5, 1.0)
- **KL-reg**: $\epsilon$ - (0.1, 1.0, 100.)
- **DAGGER**: $\alpha_{DAGGER}$ - (0, 0.5, 1.0)
- **RHPO**: Fine-tune or freeze parameters

### C.3.2 LOCOMOTION

**Learner parameters**

- Batch size: 256
- Trajectory length: 48
- Learning rate: 1e-4
- Min replay size to sample: 128
- Steps per update: 8
- Replay size: 1e6
- Target Actor update period: 25
- Target Critic update period: 100
- E-step KL constraint $\epsilon$: 0.1
- M-step KL constraints: 5e-3 (mean) and 1e-5 (covariance)

**Network parameters**

- Torso for Gaussian policies: (128, 256, 128)
- Component network size for Mixture policies: (256, 128)
- Categorical network size for MoG policies: (128,)

**Scheduler specific parameters**

- Scheduler batch size: 128
- Scheduler learning rate: 1e-4
- Min replay size to sample for scheduler: 64
- Steps per update: 4
- Available skill lengths: $K = \{n \cdot 10\}_{n=1}^{10}$
- Initial skill lengths biases: 0.95 for $k = 100$, 0.005 for the other skill lengths

**Hyper-parameter sweeps**

- $\alpha$ (MPO v/s CRR loss) - (0, 0.5, 1.0)
- **KL-reg**: $\epsilon$ - (0.1, 1.0, 100.)
- **DAGGER**: $\alpha_{DAGGER}$ - (0, 0.5, 1.0)
- **RHPO**: Fine-tune or freeze parameters

## D ADDITIONAL EXPERIMENTS

### D.1 IMPACT OF TRADE-OFF BETWEEN MPO AND CRR LOSS

**Fine-tuning with offline learning.** As discussed in Section 5 of the main text, fine-tuning often leads to sub-optimal performance since useful information can be lost early in training. In Fig. 13 we observe that fine-tuning with an offline-RL loss like CRR improves stability and learning

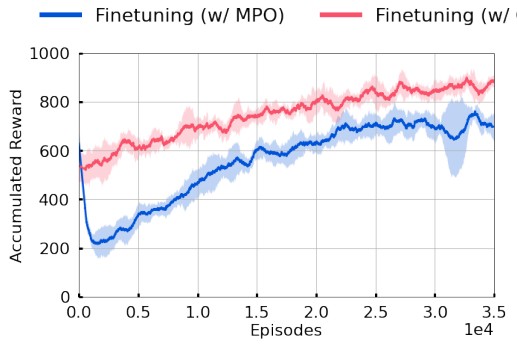
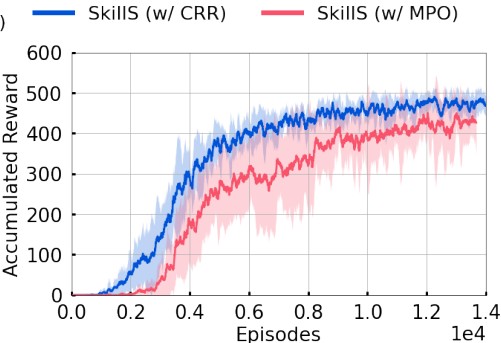

Figure 13: Comparison between regularized fine-tuning (CRR) and vanilla fine-tuning (MPO) on the GoalScoring task. Offline learning methods like CRR typically regularize towards the skill (via data in the replay) which can improve learning stability when fine-tuning.

Figure 14: Comparison when using CRR v/s MPO to infer the new skill on the Stacking task.

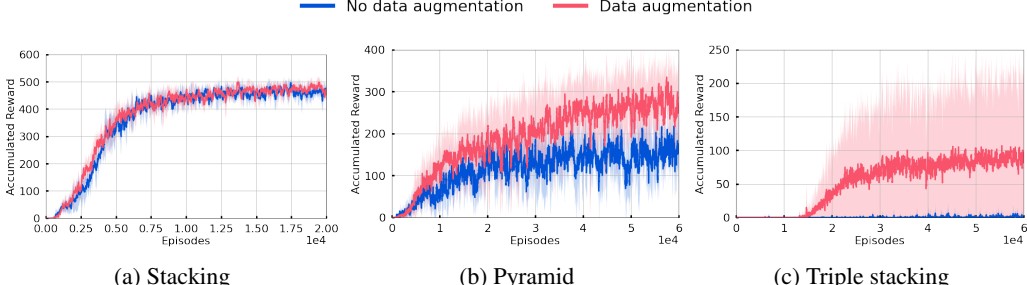

(a) Stacking          (b) Pyramid          (c) Triple stacking

Figure 15: **Data Augmentation:** Data augmentation becomes crucial when addressing harder tasks.

performance. Many offline methods, including CRR, typically regularise towards the previous skill either explicitly through the loss or implicitly via the data generated by the skill (see Tirumala et al. (2020) for a more detailed discussion on this). We hypothesize that such regularisation could help mitigate forgetting and hence improve learning. While this insight requires a more detailed analysis that is outside the scope of this work, we however have some evidence that when fine-tuning with a single skill a potentially simple algorithmic modification (from off-policy to offline RL) might already make learning more robust.

**CRR for new skills.** In Fig. 14 we compare what happens when the new skill is inferred with an offline learning algorithm (CRR) against an off-policy algorithm (MPO) on the stacking domain from Section 5. We observe that learning is improved when using an offline algorithm like CRR. For this reason, all of the experiments presented in the main text use CRR as the underlying skill inference mechanism - although the method itself is somewhat robust and agnostic to this choice. For a fair comparison, we also include a sweep using CRR as the underlying agent across when selecting the best performing baseline. However, since CRR implicitly regularises towards the skill, there is typically no effect when compared to using MPO when regularising towards a mixture.

### D.2 IMPACT OF DATA AUGMENTATION

Fig. 15 shows how data augmentation improves the performance, especially for more challenging tasks.

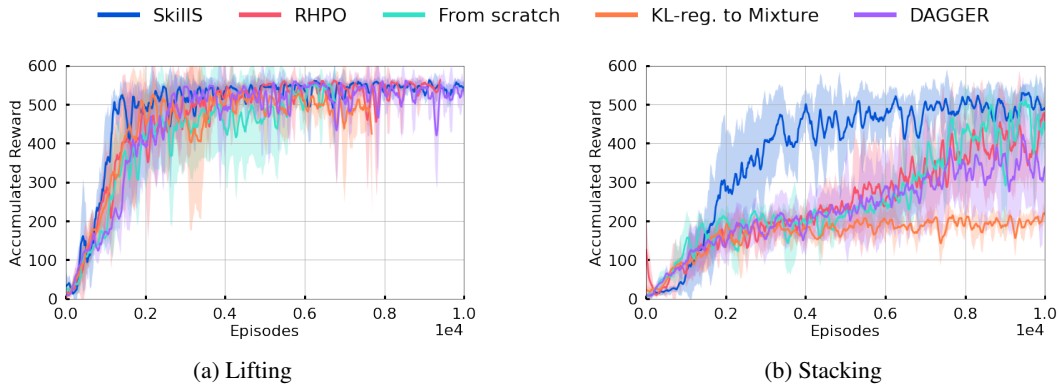

(a) Lifting

(b) Stacking

Figure 16: Performance on dense-reward manipulation tasks. With carefully-tuned shaped rewards on simpler tasks, skill reuse is not as beneficial and all methods can learn effectively.

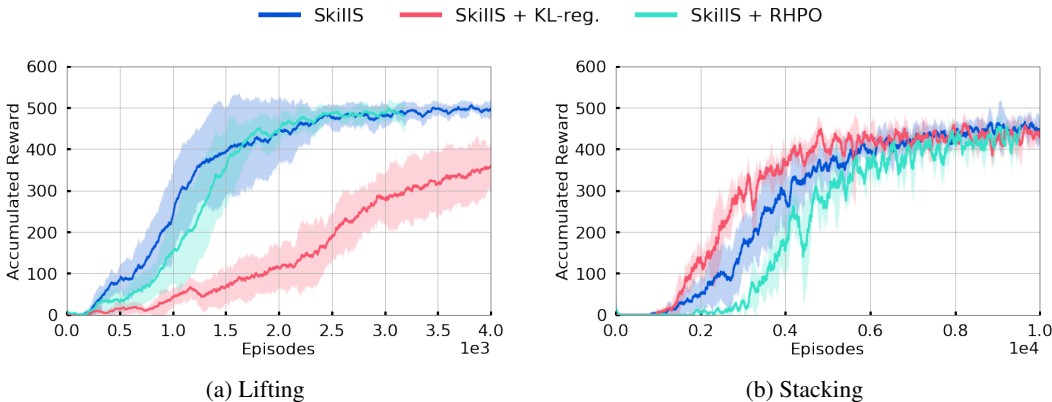

(a) Lifting

(b) Stacking

Figure 17: Performance of SkillS when combined with other approaches to transfer, showing its flexibility.

### D.3 PERFORMANCE WITH DENSE REWARDS

Fig. 16 shows the performance with all skill transfer methods on the lift and stacking tasks when the agent has access to dense rewards. We observe that, in contrast to the sparse reward results of Fig. 4, all methods learn effectively when provided with dense rewards.

Since the reward provides a useful and reliable signal to guide learning there is less need to generate good exploration data using previous skills. The advantage of skill-based methods is thus primarily in domains where defining a dense reward is challenging but the required sub-behaviors are easier to specify.

### D.4 COMBINATIONS WITH ADDITIONAL TRANSFER MECHANISMS

The separation of collect and infer processes means that SkillS can be flexibly combined with other skill techniques, included the previously analysed baselines. In Figure 17 we compare the performance of SkillS from the paper against versions of SkillS where the new skill is trained using auxiliary regularisation objectives in the manipulation Lift and Stack tasks. We found no benefit of changing the underlying algorithm to infer the new skill. The main takeaway of this result is that SkillS can flexibly combined with different algorithms to run inference which could be beneficial in some settings not considered here.

