# OpenReview forum: "SkillS: Adaptive Skill Sequencing for Efficient Temporally-Extended Exploration"
_ICLR.cc/2023/Conference — Submitted to ICLR 2023_

### Official Review · Reviewer_Qd5u · 2022-10-24

**Confidence:** 4
**Correctness:** 2
**Technical Novelty And Significance:** 2
**Empirical Novelty And Significance:** 2
**Recommendation:** 5

**Clarity, Quality, Novelty And Reproducibility:**

The clarity of the paper is good in terms of situating the method with respect to previous literature (although the related works section is a bit of an overload of citations. For example there are 10 citations to mention the idea of latent continuous space). The clarity and quality of the method is not great, as there are arbitrary choices that are not explained. Is the skill length random? It seems to be so looking later at the experiments. The section on data augmentation is completely disregarding work on intra-option learning that leverages more data.
In terms of reproducibility I have made this concern obvious, but I can't see how 5 seeds are enough for these hard exploration, difficult tasks.

**Strength And Weaknesses:**

# Strengths
- The work is well situated with respect to previous work
- The environments on which the methods are evaluated are challenging
- The authors provide a set of ablations

# Weaknesses
- The experimental details are not clear. For example, how do you leverage previous skills across seeds? How are the skills transfered in the Lift task? Why the need for adding CRC to the mix?
- The authors make multiple hand-wavy claims about how existing methods "can" fail. Any method can fail on a specific task, and saying that something "can" fail is not a great argument, as likely the new proposed approach "can" also fail.
- The experiments are done with only 5 seeds, which to me is a deal breaker. This brings a lot of questions as to the reproducibility of the results. I also couldn't find any indication of what the shaded regions represent.

**Summary Of The Paper:**

The paper tackle the problem of efficient transfer by leveraging both hierarchy and KL-regularisation. The authors claim that fine-tuning, hierarchical methods and imitation based approaches can all fail and therefore propose a method that combines elements from all of them. The authors perform experiments on robotic object-stacking tasks and gait learning and see consistent improvements.

**Summary Of The Review:**

The authors tackle an important problem, but the proposed solution is not clearly motivated and presented. The experimental details bring major concerns in terms of reproducibility.

======================================================

After discussion with the authors, I have updated my score. The main factor is the acknowledgement that reproducibility is crucial and that the authors have attempted to add more seeds to their experiments. In general the paper still seems to add many elements to the algorithm, yet some of these choices are not well motivated. For example, the authors mention that CRR significantly helps for the off-policy problem, yet looking at Figure 14 there is no significant statistical difference. This same conclusion can also be drawn by Figure 15 concerning data augmentation. Overall, this also points towards the importance of having more seeds such that the significance of these kind of choices are clearly seen.

---

> ### Author Response · Authors · 2022-11-16
> **Author rebuttal**
>
> Thank you for the detailed review and insightful comments. We have included a point by point response to the questions and concerns brought up below. The corresponding changes in the paper are marked in blue.
>
> **“The experimental details are not clear. For example, how do you leverage previous skills across seeds? How are the skills transferred in the Lift task? Why the need for adding CRC to the mix?”**
> We apologize that the details were not clear - unfortunately due to space constraints we only included the points that we felt were most pertinent to our discussion in the main text and were forced to incorporate the finer experimental details in the Appendices. In particular, Table 3 in the Appendix provides a detailed explanation of the skills used for each transfer setting including the Lift task.
> For Lift, we consider ‘Reach green’ and ‘Lift arm with closed fingers’ as useful skills and four distractor skills including reaching to other objects, opening fingers and lifting with open fingers. The rewards used to train these skills are detailed in Section C.1.4. We train one agent for each of the skills defined and then, importantly, fix the chosen skills across all the methods considered in our work for a fair comparison. Each seed considered during transfer thus uses the same underlying skill.
> While our method would work in theory with any off-policy RL algorithm, as discussed in Appendix section D.1 we found empirically that CRR performed better than MPO in some settings. Since CRR was designed for the offline RL case where the data distribution used for learning can be quite different from the policy being learnt, we posit that it is better suited to handle the highly off-policy nature of data encountered by the new skill. Please let us know if it is important for updating your score to include this information in the main paper. However, this would be challenging given the space limitations as mentioned above.
>
> **“The authors make multiple hand-wavy claims about how existing methods "can" fail. Any method can fail on a specific task, and saying that something "can" fail is not a great argument, as likely the new proposed approach "can" also fail.”**
> We demonstrate empirically that many different types of existing skill transfer methods do fail on numerous tasks in Section 5. We appreciate your comment that the claims made in the introduction were quite broad and have changed the introduction to avoid the unclear wording (e.g. “can fail”) and clarify that this is something we demonstrate empirically.
> “The experiments are done with only 5 seeds, which to me is a deal breaker. This brings a lot of questions as to the reproducibility of the results. I also couldn't find any indication of what the shaded regions represent.”
> We report performance over 5 seeds since, to the best of knowledge, this number is common practice for complex embodied RL papers due to computational constraints and the related financial costs (e.g. see Pertsch et al, 2020; Singh et al, 2021; Rao et al, 2022). We have also restarted experiments with 20 seeds to ensure robustness; unfortunately these won’t be ready before the end of the paper-editing period for all domains (given the large number of experiments and walltime requirements of up to 2 weeks) - we will add this in the next iteration of the paper. However the trends for these experiments are already consistent on 3 out of the 4 manipulation settings with what we have already plotted in the paper. We have also edited the paper to clarify that the shaded region shows one standard deviation above and below the mean over all seeds.
>
>
> **“The clarity and quality of the method is not great, as there are arbitrary choices that are not explained. Is the skill length random? It seems to be so looking later at the experiments. “**
> The set of available skill lengths (to be chosen by the scheduler) is a hyperparameter that was set empirically to work well across a number of domains - the values are specified in Appendix C.3. We also perform an ablation to demonstrate the impact of variable skill length in Figure 8 that we have updated to include more skill lengths as per Reviewer XQxL’s suggestion.
>
> **“The section on data augmentation is completely disregarding work on intra-option learning that leverages more data.”**
> We reference numerous intra-option works in the related work section (eg. Sutton et al, 1999; Bacon et al, 2017; Wulfmeier et al, 2021). Following your suggestion, we have now additionally modified the data augmentation section to reference these works as well. If you  can clarify which intra-option learning methods have been omitted, we will happily include them.

---

> > ### Author Response · Authors · 2022-11-16
> > **Author rebuttal continuation**
> >
> >
> >
> > [1] Pertsch, Karl, Youngwoon Lee, and Joseph Lim. "Accelerating reinforcement learning with learned skill priors." Conference on robot learning. PMLR, 2021.
> >
> > [2] Singh, Avi, et al. "Chaining behaviors from data with model-free reinforcement learning." Conference on Robot Learning. PMLR, 2021.
> >
> > [3] Rao, Dushyant, et al. "Learning transferable motor skills with hierarchical latent mixture policies." International Conference on Learning Representations. 2022.
> >
> > [4] Sutton, Richard S., Doina Precup, and Satinder Singh. "Between MDPs and semi-MDPs: A framework for temporal abstraction in reinforcement learning." Artificial intelligence 112.1-2 (1999): 181-211.
> >
> > [5] Bacon, Pierre-Luc, Jean Harb, and Doina Precup. "The option-critic architecture." Proceedings of the AAAI Conference on Artificial Intelligence. Vol. 31. No. 1. 2017.
> >
> > [6] Wulfmeier, Markus, et al. "Data-efficient hindsight off-policy option learning." International Conference on Machine Learning. PMLR, 2021.

---

> > ### Comment · Reviewer_Qd5u · 2022-11-17
> > **Response to the Author rebuttal**
> >
> > I would like to thank the authors for their rebuttal.
> >
> > I appreciate the initiative to run more seeds, as the argument that other researchers have also only ran 5 seeds is not great given the current state of reproducibility in reinforcement learning. Hopefully the authors can update the paper with the results they have gathered.
> >
> > From what I understand, every seed uses the same underlying skill. I am not sure if this is the right way to do transfer, it feels like to obtain better statistical significance we would want to have the same type of skill being learned by N learners (where N is the number of seeds) such that each seed would start truly randomly. By comparison with the online transfer setting, where we don't get to select from which seed we transfer, this is more appropriate.
> >
> > I am still not convinced by the use of CRR. Do the baselines also use this method? It would be very important to make an ablation on this choice that seems arbitrary.

---

> > > ### Author Response · Authors · 2022-11-18
> > > **Response to reviewer rebuttal**
> > >
> > > Thank you very much for the quick response. We agree that reproducibility is a challenge in reinforcement learning and describe the reasons and constraints underlying our choices below.
> > >
> > > **I appreciate the initiative to run more seeds, as the argument that other researchers have also only ran 5 seeds is not great given the current state of reproducibility in reinforcement learning.**
> > >
> > > A high number of seeds is definitely helpful when evaluating RL agents and we appreciate your comments to strive for this. We are currently running these experiments and have already replaced a share of our figures. Please have a look. In general, every experiment creates additional costs. For any work, there is a trade-off between these costs and adding more evidence for an existing experiment. In our case, we are running a total of 720 experiments for up to 2 weeks, leading to a cost of 1680 GPU weeks.
> > >
> > > Many papers, including ours, often add additional domains,  to provide further insights. This requires trading off the number of seeds per domain with the number of domains. We added multiple domains and ablations to aggregate insights across experiments. Due to the computational cost associated with these, we are limited in the number of experiments. In our case, the choice was also driven by the fact that on most domains not a lot of variance was observed over the 5 seeds; such that additional seeds were unlikely to change any outcomes. Our latest results seem to be consistent with this rationale.
> > >
> > > **Hopefully the authors can update the paper with the results they have gathered.**
> > >
> > > We have updated the plots to include all the results that we managed to run and have converged so far. While we will have all the plots for a camera-ready version if the paper is accepted, unfortunately, we are unlikely to have all of them within the rebuttal deadline, due to the computational cost we mention above.
> > >
> > > **From what I understand, every seed uses the same underlying skill. I am not sure if this is the right way to do transfer, it feels like to obtain better statistical significance we would want to have the same type of skill being learned by N learners (where N is the number of seeds) such that each seed would start truly randomly. By comparison with the online transfer setting, where we don't get to select from which seed we transfer, this is more appropriate.**
> > >
> > > This is a good point, thanks for raising it. We indeed have a similar analysis in Section 5.2.2 where we ablate the robustness of our method with respect to the quality and number skills. In fact, when we use multiple skills of the same nature (i.e. “many distractors” or “many useful skills” in Fig. 6) we indeed use multiple “distractor” and “useful” skills learned for the same tasks but from different seeds (3 in our experiments). We do recognize that this was not clearly stated in the main paper nor in the Appendix. We have now made the relevant changes in the paper.
> > >
> > > **I am still not convinced by the use of CRR. Do the baselines also use this method? It would be very important to make an ablation on this choice that seems arbitrary.**
> > >
> > > We have in fact run ablations for all of our baselines using off-policy and offline RL updates. We found across the board that CRR works better in this setting and have included an ablation in the original submission (Appendix D.1 which is referred to in the main text). For all of the baselines that were considered we included a hyper-parameter \alpha that trades off the baseline loss against a CRR loss. In other words, setting \alpha=1 is pure CRR while \alpha=0 is just the baseline loss. We swept over values of (0, 0.5, 1.0) for the \alpha for each algorithm considered and plotted the best results for each. We have updated the paper to more clearly state this in the Appendix and hope this clarifies the reviewer’s concern.
> > >
> > >
> > > We hope that these comments provide sufficient background for decisions and trade-offs made. We hope that the revised draft with more experiments is sufficiently better to improve your score. Please do reach out if there are other aspects of our work that can be clarified further.

---

> ### Author Response · Authors · 2022-12-09
> **Reply to Updated comment from Reviewer**
>
> Thank you for your comments and for updating your score after our response. Regarding the motivation of CRR and data augmentation in our algorithm - we would like to highlight that these are merely algorithmic choices that moderately improve performance are  orthogonal to the primary message of our work. This is also why we chose to include these results in the Appendix and have not included them in the list of key contributions for our work. We believe these are valid choices because:
>
> 1) For CRR - we sweep over the \alpha from pure MPO to pure CRR and report the best result. Since all our baselines also include this sweep, we believe this is a fair comparison and report the best results.
>
> 2) While the reviewer’s observation that CRR only modestly improves performance in Figure 14 is fair, since we are reporting the best results averaged across seeds, we chose to report the CRR version of the experiment. We would like to highlight that the standard deviation of the curves in this figure is not very large indicating that adding more seeds is unlikely to change the trend.
>
> 3) The data augmentation in Figure 15 does not show much performance improvement in the Stacking or Pyramid tasks but shows a drastic improvement of performance in the Triple stacking domain (rightmost figure) which is the most complex task considered in this setting. This indicates that the data augmentation could be particularly useful as we try to tackle harder settings.
>
> We hope this allays some of the fears raised by the reviewer and hope they will take this into consideration in the final days left for discussion.

---

### Official Review · Reviewer_C7sy · 2022-10-24

**Confidence:** 3
**Clarity, Quality, Novelty And Reproducibility:** This paper is well written. And the p…
**Correctness:** 3
**Technical Novelty And Significance:** 3
**Empirical Novelty And Significance:** 3
**Recommendation:** 6

**Strength And Weaknesses:**

Strength: The idea of learning new policy with data generated by following sequencing policies makes sense to me. Especially for complex tasks that can be devide into subtasks, the proposed method will lead to an efficient learning strategy which gradually learn better about the entire task. Also the proposed method is shown to be significantly better than others on sparse-reward manipulation tasks. The sparsity of the reward can be handled by gradually learning subtasks.

Weakness: There is no convergence analysis for the proposed HCMPO algorithm. Without any assumption on the continuity of the subtasks, it is not clear that the proposed algorithm is guaranteed  to converge to the optimal policy. It will be better if the authors can provide certain analysis on this.

**Summary Of The Paper:**

This paper considers the problem of transferring learned policies to a new target task sequentially. A new method SkillS is proposed for the problem which separates the data collection from solution inference. During data collection phase, a high-level schedule chooses the best learned skill to maximize the task reward obtained from the environment. The scheduler is trained with a proposed algorithm, Hierarchical Categorical MPO. The task solution is learned via an off-policy learning algorithm that optimists a new skill for the given task. The scheduler can learn to sequence temporally-extended skills and bias exploration towards useful regions of trajectory space, and the new skill can then learn an optimal task solution off-policy. Through experiments, the authors show the proposed method outperforms related method across all sequential tasks.

**Summary Of The Review:**

The idea in this paper makes sense to me. The proposed algorithm is shown to perform significantly better than other on certain tasks.

---

> ### Author Response · Authors · 2022-11-16
> **Author rebuttal**
>
> Thank you for the constructive review. We have included clarifications and a point by point response to any open questions. The corresponding changes in the paper are marked in blue.
>
> We would like to start by clarifying that the two processes happen in parallel. Both scheduler and new skill are being trained at the same time such that the learning process of either directly affects the other. We have additionally emphasized this point in the revised paper to improve clarity.
>
> **“There is no convergence analysis for the proposed HCMPO algorithm. Without any assumption on the continuity of the subtasks, it is not clear that the proposed algorithm is guaranteed to converge to the optimal policy. It will be better if the authors can provide certain analysis on this.”**
> We agree on the importance of theoretically grounding empirical results. The focus of this paper lies on a broad empirical analysis. This includes the relations between task and reloaded skill sets (i.e. skills which are more and less aligned with solving a particular task), number of skills and further conditions including the impact of temporal abstraction. An exact convergence analysis for an algorithm with two components learning in parallel (scheduler and new skill) and relying on deep neural networks is hard if not impossible to obtain unless major simplifying assumptions are made. Typically this results in analysis in simpler domains with discrete state and action spaces without assuming function approximation with neural networks. This is the reason why this type of analysis is generally uncommon across recent work in deep RL in particular for more complex domains as the ones used in this work.
>
> We could further include a more detailed response and can adapt the paper if you can clarify the terminology regarding ‘continuity of subtasks’.
>
> Please do not hesitate to ask for any clarifications.

---

### Official Review · Reviewer_XQxL · 2022-10-25

**Confidence:** 4
**Correctness:** 3
**Technical Novelty And Significance:** 3
**Empirical Novelty And Significance:** 4
**Recommendation:** 8

**Clarity, Quality, Novelty And Reproducibility:**

* This paper is generally clear and of high quality, except for a few minor points raised in the above section.

* The method and analysis appear to be novel.

* There is no reproducibility statement or promise to publish code, but the experimental details in the appendix appear to be comprehensive, and the main ideas in the work are simple enough that they should be able to be applied by future work even without the precise implementation details.

**Strength And Weaknesses:**

Strengths:

* The exposition is detailed and clear.

* The problem is well defined and compelling and the resulting method is well-motivated. The discussion of prior work is comprehensive, and is presented in a way that effectively feeds into the motivation, and even makes the new method seem inevitable.

* The design decisions are generally well-supported, are described clearly and concisely in the main body (and in even more detail in the Appendix).

* The environments chosen for evaluation are challenging and diverse.

* The experimental results are generally compelling and comprehensive. The expected ablations and comparisons are performed.

* Useful avenues for future work are discussed in the conclusion.

Weaknesses/questions:

* None major.

* How is the dependence on the frozen skills reduced over the course of training, if the scheduler is encouraged to maximize reward? Does this simply occur naturally? This is how I currently understand it.

* Related to the previous question: did the authors try to regularize learning in any way to encourage the scheduler to choose the new skill more often? Or did this always naturally emerge? I would imagine issues with this not encouraging this; e.g. the scheduler rarely choosing the new skill in certain situations (relying on frozen skills in these), and then come evaluation, the new skill will perform poorly in these cases. Is there any intuition for why this does not (seem to) occur?

* The results seem to demonstrate that variable temporal abstraction is necessary for best performance, which is an interesting finding. However, the authors appear to compare against only one constant for the fixed temporal abstraction results. How was 200 steps chosen? Can we see results with multiple other values to be convinced that the variable choice matters, and not that the fixed value is merely a poor choice? It could be helpful to compare the variable SkillS to the fixed temporal abstraction SkillS with the constant set to be the average skill length discovered by variable SkillS. This would more convincingly demonstrate the benefit of variable length skills. With my understanding of the current results, I’m not convinced.

* Figure 7a: A little confused by this graph. Are these curves generated using different manually specified schedulers that were otherwise trained with the same learning procedure (i.e. “New Skill” is the performance of the scheduler only choosing the new skill after being trained the same way as the other too)? This seems to be the case, but either way, readers may benefit from a clearer explanation of this.

**Summary Of The Paper:**

This paper proposes SkillS, a method for reusing skills in new reinforcement learning tasks that employs a learned high-level scheduler to execute skills, and a learned new skill, which is included in the set the scheduler selects. In this way, the benefit of temporally extended exploration is retained by executing previously learned skills for numerous steps at a time, catastrophic forgetting is avoided by learning an entirely new set of parameters separate from the frozen skills, and the potential suboptimal convergence of HRL is avoided by allowing the new skill to be unconstrained by previously learned behaviors. They suitably liken their approach to related works that perform “transfer via data” (i.e. the frozen skills are not used in final evaluation, but rather to generate useful training data for the new unconstrained skill). The scheduler is trained via a discretized version of MPO, and the new skill is learned using CRR. In a number of robotic manipulation and locomotion domains, the authors empirically demonstrate the performance gains achieved by SkillS relative to relevant approaches in the literature. The authors also include additional results, including a demonstration of SkillS’ ability to learn with different quantities/quality of skills, analysis of the selection trends of the scheduler, analysis of the utility of flexible temporal abstraction versus fixed, and a study of the importance of having separate collect/infer mechanisms.

**Summary Of The Review:**

* The paper is well-written and detailed, the method is well-motivated and grounded in relevant prior work, and the results are generally compelling and comprehensive. There are a few minor concerns/questions I have for the authors, but otherwise I am in favor of recommending this paper for acceptance, as I believe it will benefit the research community.

---

> ### Author Response · Authors · 2022-11-16
> **Author rebuttal**
>
> Thank you for the detailed review and insightful comments. We have included a point by point response to the questions and concerns brought up below. The corresponding changes in the paper are marked in blue.
>
> **“How is the dependence on the frozen skills reduced over the course of training, if the scheduler is encouraged to maximize reward? Does this simply occur naturally? This is how I currently understand it. Related to the previous question: did the authors try to regularize learning in any way to encourage the scheduler to choose the new skill more often? Or did this always naturally emerge? I would imagine issues with this not encouraging this; e.g. the scheduler rarely choosing the new skill in certain situations (relying on frozen skills in these), and then come evaluation, the new skill will perform poorly in these cases. Is there any intuition for why this does not (seem to) occur?.”**
> Exactly as you suggest, the scheduler tends to naturally favor the new skill over frozen skills over the course of training; there is no explicit objective to incentivize this. In general we observed the following training dynamics: the scheduler chooses the frozen skills (with our pre-defined initial bias to choose longer skill lengths) since they lead to higher reward early on; while the new skill learns the task solution with an offline RL algorithm; and eventually the scheduler picks the new skill most often. This is shown in Figure 5c in the paper. As long as there are rewarding trajectories being generated by the scheduler, we empirically found that the new skill eventually converges to a good solution even if it is not often picked. One way to think of this is that using an offline RL algorithm effectively enables learning from highly off-policy data. In other words the new skill learns from the best of the replay data, while being able to differ from skill behavior to reduce the dependence on reloaded frozen skills.
>
> **“The results seem to demonstrate that variable temporal abstraction is necessary for best performance, which is an interesting finding. However, the authors appear to compare against only one constant for the fixed temporal abstraction results. How was 200 steps chosen? Can we see results with multiple other values to be convinced that the variable choice matters, and not that the fixed value is merely a poor choice? It could be helpful to compare the variable SkillS to the fixed temporal abstraction SkillS with the constant set to be the average skill length discovered by variable SkillS. This would more convincingly demonstrate the benefit of variable length skills. With my understanding of the current results, I’m not convinced.”**
> This is a very good point for the ablation - thank you for the suggestion! We are re-running this ablation with multiple choices for the timestep length including [50, 100, 150]. We found that the average skill length discovered by SkillS varies based on the domain and transfer setting but usually is in the range of 50 to 100 so this ablation should help shed more light on the benefit of variable skill length. In our updated Figure 8 in the paper (marked with a blue boundary), we observe that the flexible temporal abstraction is particularly useful for the hardest task of Triple Stacking. For the other tasks, there is no single skill length which is optimal and the best choice varies by task. In contrast, the values for the flexible temporal abstraction were chosen once and consistently work well across all domains.
>
>  **“Figure 7a: A little confused by this graph. Are these curves generated using different manually specified schedulers that were otherwise trained with the same learning procedure (i.e. “New Skill” is the performance of the scheduler only choosing the new skill after being trained the same way as the other too)? This seems to be the case, but either way, readers may benefit from a clearer explanation of this.”**
> When training the SkillS algorithm we can plot the performance of the scheduler which contains the new skill and the frozen skills, or, just the new skill which is trained offline. In Figure 7a, ‘New Skill’ refers to the performance of the new skill and ‘Scheduler (incl. new skill)’ refers to the performance of the corresponding scheduler. We also ran a separate experiment with a scheduler that only has access to the frozen skills (and no new skill). We refer to this as ‘Scheduler (without new skill)’ in the figure. We observe that having access to a new skill that can learn to deviate from the frozen skills is critical for SkillS to perform well. We hope this clarifies the figure and have updated the text to provide a clearer explanation of it.

---

### Official Review · Reviewer_E1qM · 2022-11-05

**Confidence:** 4
**Correctness:** 2
**Technical Novelty And Significance:** 2
**Empirical Novelty And Significance:** 2
**Recommendation:** 3

**Clarity, Quality, Novelty And Reproducibility:**

Clarity: The paper is generally clear. It is well organised.

Originality: To the best of my knowledge, the paper is novel in the particular approach being proposed. That said, I would not rate the paper as particularly innovative. Using pre-defined skills without modification is a standard use of the options framework.

Quality: The analysis in the paper is informative. However, it is essentially aiming to empirically show that the approach **can** be useful. But, in many cases, the approach will not be particularly useful, and can even be harmful. The results in the paper are not particularly informative towards understanding how useful the approach will be generally and the conditions under which it is likely to be useful.



**Strength And Weaknesses:**

The primary strength of the paper is that it explores the important problem of knowledge transfer between tasks. In addition, the approach taken is sensible. However, I cannot claim that it is fundamentally innovative. The primary weakness is that the utility of the approach will depend on the particular task and the particular skills that are being used. While the paper gives examples of its successful use, it is not difficult to think about many cases in which the approach will not be successful. The paper does not present useful theory or analysis on the conditions under which the approach is likely to succeed.

**Summary Of The Paper:**

The paper explores an approach to using existing (learned) skills in a new task in the same domain. The approach keeps the existing skills intact (i.e., does not modify them); learns a policy over them (called the "scheduler") that maximises current task reward; and, at the same time, learns another policy (called the "new skill") that also maximises current task reward but by only using the primitive actions in the domain. Action selection in the domain is done by the scheduler. The approach is tested in two simulated robotics domains.

**Summary Of The Review:**

The paper presents an approach to knowledge transfer between tasks. While the approach is sensible, and will produce good results in some cases, the results in the paper are not particularly informative towards understanding how useful the approach will be generally and the conditions under which the approach is likely to be useful.

---

> ### Author Response · Authors · 2022-11-16
> **Author rebuttal**
>
> Thank you for the valuable feedback. We have addressed the key concerns below. The corresponding changes in the paper are marked in blue.
>
> **“ The primary weakness is that the utility of the approach will depend on the particular task and the particular skills that are being used. “**
> Thanks for the comment; we agree that any approach to skill-based transfer will depend on the nature of the method and the tasks chosen. In this work, we have focused on embodied settings in which we have related skills available, and need to transfer in a sample-efficient way to a range of related tasks (some of which may be compositional in the existing skills). While some experiments assume the skills are relevant, we also run ablations which show the robustness of the method in the presence of varying quantities of distractor skills (Figure 6), over numerous tasks. We observe that even in the presence of distractor skills like reaching to a red object when trying to lift a blue one, our method is quite robust. We have modified the introduction to make the scope (of embodied skills) clear. Finally, it is worth highlighting that the dependence to the available skills is reduced with respect to common hierarchical approaches (e.g. options framework) as the new policy, i.e. the solution for the task at hand, is separately learned from the collected data.
>
> **"That said, I would not rate the paper as particularly innovative. Using pre-defined skills without modification is a standard use of the options framework."**
> Thank you for raising this point, and we hope this is clarified in the new revision of the paper. Using pre-defined frozen skills is indeed a standard use of the options framework (and in fact, the approach of many skill-based methods we discuss in our related work). However, one of the key contributions here is our inspiration from the “collect and infer” paradigm, which separates the processes of exploration / data collection; and inference / policy learning. With this property, our method can make use of frozen skills (and the new policy) to perform exploration, without constraining the final policy, which in previous work is often defined over the space of skills. The plot of the “Scheduler (incl. new skill)” in Fig 7a shows how frozen skills enable fast exploration, as significant reward is experienced early in training. However, they also constrain the final performance that “Scheduler (incl. new skill)” eventually achieves. The same plot shows how instead the “New policy” overtakes the final scheduler performance, being learned from the data and without incorporating the existing skills. We have modified the introduction to further emphasise this contribution and contextualize our work.

---

### Author Response · Authors · 2022-11-16
**Author rebuttal - General points**

Thank you for the valuable feedback. We have included individual feedback to address specific concerns and include a summary of high level changes here:
- We are running experiments for the main results of the paper (Figures 4, 5a and 5b) with 20 seeds. Since some experiments take a while to run (up to two weeks), we will continue to update the figures as we approach the deadline. However the results that we already have (highlighted in blue) are consistent with our previous results and show that SkillS performs far better than any baselines in the sparse reward settings considered.
- We have updated Figure 8 to include a comparison against a larger set of skill lengths to illustrate the benefit of the temporal flexibility. As mentioned above, we will keep updating the figures as we approach the deadline.
- We have made a number of improvements to the text based on the feedback which are highlighted in blue for convenience.

We are happy to make further changes to the paper before the deadline and look forward to continuing ongoing discussions. While we will strive to include the latest figures, we would like to stress the computational cost of re-running these experiments is quite onerous and all results may not have converged by the deadline.

---

> ### Author Response · Authors · 2022-12-09
> **Reply to all reviewers**
>
> Thank you for the reviews and feedback. We are hoping that the reviewers who have not yet replied might consider updating their score. If not, we are happy to address any lingering concerns and questions you may have before the end of the rebuttal.

---

### Decision · Program_Chairs · 2023-01-20

**Decision:**

Reject

**Justification For Why Not Higher Score:**

The contribution of this paper is empirical, but the lack of motivation for some choices and the low significance of some experiments put this paper slightly below the bar.

**Justification For Why Not Lower Score:**

N/A

**Metareview: Summary, Strengths And Weaknesses:**

This paper proposes a method for reusing temporally-extended skills in new reinforcement learning tasks.
The proposed approach addresses some of the issues affecting related works.
The effectiveness of the proposed solution is demonstrated by empirical validation using robotic manipulation and locomotion domains.
After reading each others' reviews and the authors' feedback, the reviewers discussed their opinions about this paper, without reaching a consensus.
Although the authors' answers effectively solved some of the issues raised by the reviewers, they still complain that the proposed approach contains many choices that are not well-motivated and whose utility (in some cases) does not seem to be significant.
As the approach is not supported by a theoretical analysis, it is very important to have relevant empirical results that help in advance our comprehension of the transfer of knowledge among reinforcement learning tasks.
We encourage the authors to consider reviewers' suggestions while preparing a new version of their paper.